# PopulAtion Parameter Averaging (PAPA)

**Alexia Jolicoeur-Martineau**                    *alexia.j@samsung.com*
*Samsung - SAIT AI Lab, Montreal*

**Emy Gervais**                                   *emy.gervais0@gmail.com*
*Independent*

**Kilian Fatras**                                 *kilian.fatras@mila.quebec*
*Mila, McGill University*

**Yan Zhang**                                     *y2.zhang@samsung.com*
*Samsung - SAIT AI Lab, Montreal*

**Simon Lacoste-Julien**                          *simon.lj@samsung.com*
*Mila, University of Montreal*
*Samsung - SAIT AI Lab, Montreal*
*Canada CIFAR AI Chair*

**Reviewed on OpenReview:** *https: // openreview. net/ forum? id= cPDVjsOytS*

## Abstract

Ensemble methods combine the predictions of multiple models to improve performance, but they require significantly higher computation costs at inference time. To avoid these costs, multiple neural networks can be combined into one by averaging their weights. However, this usually performs significantly worse than ensembling. Weight averaging is only beneficial when different enough to benefit from combining them, but similar enough to average well. Based on this idea, we propose *PopulAtion Parameter Averaging* (PAPA): a method that combines the **generality of ensembling** with the **efficiency of weight averaging**. PAPA leverages a population of diverse models (trained on different data orders, augmentations, and regularizations) while slowly pushing the weights of the networks toward the population average of the weights. We also propose PAPA variants (PAPA-all, and PAPA-2) that average weights rarely rather than continuously; all methods increase generalization, but PAPA tends to perform best. PAPA reduces the performance gap between averaging and ensembling, increasing the average accuracy of a population of models by up to 0.8% on CIFAR-10, 1.9% on CIFAR-100, and 1.6% on ImageNet when compared to training independent (non-averaged) models.

## 1 Introduction

Ensemble methods (Opitz and Maclin, 1999; Polikar, 2006; Rokach, 2010; Vanderheyden and Priestley, 2018) leverage multiple pre-trained models for improved performance by taking advantage of the different representations learned by each model. The simplest form of ensembling is to average the predictions (or logits) across all models. Although powerful, ensemble methods come at a high computational cost at inference time for neural networks due to the need to store multiple models and run one forward pass through

every network. Ensembles are especially problematic on mobile devices where low latency is needed (Howard et al., 2017) or when dealing with enormous networks such as GPT-3 (Brown et al., 2020).

An alternative way of leveraging multiple networks is to combine various models into one through weight averaging (Wortsman et al., 2022). This method is much less expensive than ensembling. However, there is usually no guarantee that weights of two neural networks average well by default (Ainsworth et al., 2023). It is only possible to average the weights directly in special cases, such as when the multiple snapshots of the same model are taken (Izmailov et al., 2018), or when a model is fine-tuned with similar data for a small number of steps (Wortsman et al., 2022). In general, starting with the same weight initialization across models is not enough to expect good performance after averaging (Frankle et al., 2020).

Permutation alignment techniques (Tatro et al., 2020; Singh and Jaggi, 2020; Entezari et al., 2022; Ainsworth et al., 2023; Peña et al., 2022) were devised to minimize the decrease in performance after interpolating between the weights of two networks. Other recently proposed techniques, such as REPAIR and greedy model soups, also exist to improve performance after interpolation/averaging. REnormalizing Permuted Activations for Interpolation Repair (REPAIR) (Jordan et al., 2023) mitigate the variance collapse through rescaling the preactivations. Meanwhile, greedy model soups (Wortsman et al., 2022) and DiWA (Rame et al., 2022) choose a subset of models so that averaging works well.

Given the recent developments of techniques to improve the performance of weight averaging, we explore the idea of weight averaging to get the benefits of ensembling in a single model. With this goal in mind, we make the following contributions:

1. We propose PopulAtion Parameter Averaging (PAPA) (Section 2) as a simple way of leveraging a population of networks through averaging for better generalization. In PAPA, multiple models are trained independently on slight variations of the data (random data orderings, augmentations, and regularizations), and every few stochastic gradient descent (SGD) steps, the weights of each model are pushed slightly toward the population average of the weights. We also return the model soups at the end of training to obtain a single model.

2. We propose PAPA variants that are more amenable to parallelization, where the weights of each model are rarely replaced every few epochs by i) the average weights of all models (PAPA-all) or ii) the average weights of two randomly selected models (PAPA-2).

3. We demonstrate in Section 4 that PAPA and its variants lead to substantial performance gains when training small network populations (2-10 networks) from scratch with low compute (1 GPU). Our method increases the average accuracy of the population by up to 0.8% on CIFAR-10 (5-10 networks), 1.9% on CIFAR-100 (5-10 networks), and 1.6% on ImageNet (2-3 networks).

## 2  PopulAtion Parameter Averaging (PAPA)

In this section, we describe PopulAtion Parameter Averaging (PAPA), a simple and easy-to-use method to leverage a population of models through averaging in order to gain the benefits of ensembling in a single model. We then describe how to handle changes in learning rate. Afterwards, we discuss how to perform efficient inference with the population of networks. Finally, we present two special cases of our method called PAPA-all and PAPA-2. Figure 1 shows an illustration of PAPA and Algorithm 1 provides the full description of PAPA and its variants (PAPA-all and PAPA-2). In the following, we elaborate on the details of PAPA.

### 2.1  Training a population of networks by pushing toward the average (PAPA)

As previously mentioned, the idea of averaging the weights of networks is simple but not always trivial to perform. In particular, it usually requires model weights to be aligned in terms of weight permutations (Ainsworth et al., 2023). Our solution is to move the weights far enough from each other to bring diversity, but not too far apart so that they do not become too dissimilar (which leads to bad performance after averaging); applying this idea during training is what constitutes PAPA.

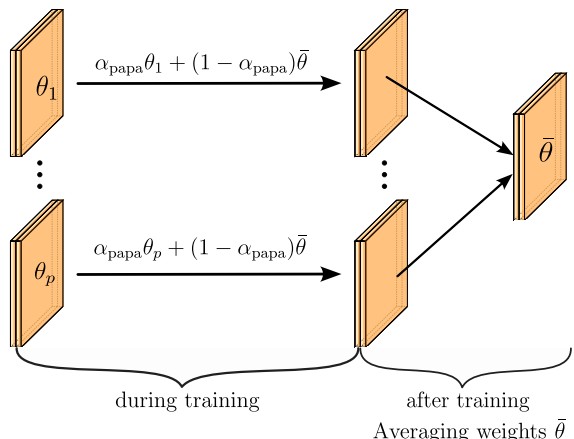

Figure 1: Illustration of PAPA. Multiple networks (with weights $\theta_j$) are trained on slight variations of the dataset. Every few (10) iterations, the weights are pushed slightly toward the population average of the weights $\bar{\theta} = \sum_{j=1}^{p} \theta_j$. After training, the weights are averaged to get a single network.

---

**Algorithm 1:** PAPA

---

**Input:** training dataset $D$, evaluation (or training) dataset $D'$, total epochs $n_{\mathrm{epochs}}$, learning rate (lr) $\gamma$, learning rate schedule $S$, averaging frequency $f$ (in number of SGD steps), EMA-rate for PAPA $\alpha_{\mathrm{papa}}$, population size $p$, set of data augmentations and regularizations $\pi = \{\pi_j\}_{j=1}^{p}$, $k$ REPAIR iterations, PAPA-2 = false, PAPA-all = false (otherwise, PAPA is used);

Initialize population $\Theta \leftarrow \{\theta_1, \cdots, \theta_p\}$;

Initialize training data $\mathcal{D} \leftarrow \{\mathrm{shuffle}(D), \cdots, \mathrm{shuffle}(D)\}$;

$n_{\mathrm{total}} \leftarrow 0$;

$\gamma_0 \leftarrow \gamma$; // `initial learning rate`

**for** $n = 1 : n_{epochs}$ **do**

    **for** $i = 1 : n_{iterations}$ **do**

        **for** $j = 1 : p$ **do**

            Sample the $i$-th mini-batch of data from $\mathcal{D}_j$;

            Data augment the data using $\pi_j$ (Optional);

            Do a forward and backward pass;

            Regularize the output using $\pi_j$ (Optional);

            Update $\theta_j$ using the optimizer (SGD, Adam, etc.);

        **end**

        Update $\gamma$ based on learning rate schedule $S$

        $n_{\mathrm{total}} \leftarrow n_{\mathrm{total}} + 1$

        **if** $n_{total}$ is divisible by $f$ **then**

        $\bar{\Theta} \leftarrow \mathrm{Averaging}(\Theta, m = 2$ if PAPA-2 else $p)$;

        **if** PAPA-2 or PAPA-all **then**   $\bar{\Theta} \leftarrow \mathrm{REPAIR}(\bar{\Theta}, D, \pi, k)$ // `optional`

        **else**   $1 - \alpha'_{\mathrm{papa}} \leftarrow \frac{\gamma}{\gamma_0}(1 - \alpha_{\mathrm{papa}})$; // `scales proportionally to learning rate` $\gamma$

        **for** $j = 1 : p$ **do**   $\theta_j \leftarrow \alpha'_{\mathrm{papa}}\theta_j + (1 - \alpha'_{\mathrm{papa}})\bar{\theta}_j$;

        $n_{\mathrm{total}} \leftarrow 0$

    **end**

**end**

$\bar{\theta} \leftarrow \frac{1}{p}\sum_{j=1}^{p}\theta_j$; // `average soup`

$\hat{\theta} \leftarrow \mathrm{GreedySoup}(\Theta, D')$; // `greedy soup`

**return** $\bar{\theta}, \hat{\theta}$;

---

To obtain a diverse and well-behaving set of weights, we start with a population of $p$ models with random initializations, each having its own data order, data augmentations, and regularizations. To keep the weights close enough for good averaging, at every few stochastic gradient descent (SGD) optimizer steps, we slowly

push the weights toward the population average of the weights using an exponential moving average (EMA) as follows:

$$\theta_j^{(i)} \leftarrow \alpha_{\text{papa}}\theta_j^{(i)} + (1 - \alpha_{\text{papa}})\bar{\theta}^{(i)}, \tag{1}$$

where $\alpha_{\text{papa}}$ is the EMA rate, $\theta_j^{(i)}$ the weight of the $j$-th network at the $i$-th optimization step, and $\bar{\theta}^{(i)}$ the population average of the weights at the $i$-th optimization step.

We use a small enough EMA rate so that the models remain well-aligned but large enough so that the models remain diverse. Since applying the EMA at every step would be costly, we apply it at every 10 steps to amortize its cost (making its cost fully negligible). In our experiments, we use $\alpha_{\text{papa}} = 0.99$ when training from scratch and $\alpha_{\text{papa}} = 0.9995$ when fine-tuning. See Table 17 for an ablation on the values of $\alpha_{\text{papa}}$.

## 2.2 Special cases of PAPA when averaging rarely instead of frequently (PAPA-all & PAPA-2)

When $\alpha_{\text{papa}} = 0$, instead of pushing the weights toward the population average, we are replacing the weights with the population average. Furthermore, if we were to amortize the case where $\alpha_{\text{papa}} = 0.999$ at every iteration on a very large number of steps, this would effectively lead to $\alpha_{\text{papa}} \approx 0$ every few epochs; for example, on CIFAR-10 (Krizhevsky et al., 2009), when averaging every 5 epochs using a mini-batch of 64, we have that $\alpha_{\text{papa}} = .999^{(781 \text{ iterations} \times 5 \text{ epochs})} \approx 0.02$.

To explore the case where we replace the weights by the population average every few epochs, we consider two ways of averaging the models: 1) averaging all models to make a single average model that replaces every member of the population (PAPA-all), and 2) every member of the population is replaced by averages of different random pairs of models (PAPA-2). The main advantage of these methods over PAPA is that they are more amenable to parallelization; the overhead cost of communicating the weights between multiple GPUs can be considerable, but this cost is amortized when averaging rarely. In practice, for small dataset (e.g., CIFAR-10, CIFAR-100), we found that averaging frequently (every 5-10 epochs) worked best when training from scratch and more sparsely (every 20 epochs) when fine-tuning. For large dataset (e.g., ImageNet), we averaged every epoch. See Table 15 for an ablation on the frequency of averaging.

Models that evolve separately for long periods may become wildly different, making averaging more difficult (which we observe in the first epochs of training). Thus, as an optional step, we use REPAIR (Jordan et al., 2023), a method that mitigates variance collapse after interpolating weights, to increase the performance of the averaged model; this leads to a minor but consistent increase in performance. See Section A.2 for more details.

## 2.3 Handling changes in learning rates

Standard training protocols for deep learning often use decreasing learning rate schedules. When the learning rate changes over time, the trade-off between the SGD optimizer step and PAPA also changes. As the learning rate $\gamma$ goes down, exploration goes down, while the effect of pushing toward the average remains the same; this makes it harder for the models to gain enough diversity to benefit from averaging. This can be observed in the following definition:

**Definition 2.1.** Let $L$ be the loss function, $\theta_j$ be the weights of the $j$-th neural network, $\alpha_{\text{papa}}$ be the update PAPA parameter and $\gamma$ be the learning rate. The update of our method PAPA reads

$$\theta_j \leftarrow \alpha_{\text{papa}}\theta_j + \gamma\Big(-\alpha_{\text{papa}}\nabla_\theta L(\theta_j) + \frac{1 - \alpha_{\text{papa}}}{\gamma}\bar{\theta}\Big).$$

The quantity $(1 - \alpha_{\text{papa}})/\gamma$ is influenced by changes in learning rate during training. As $\gamma$ decreases, the effect of PAPA becomes disproportionately large compared to the SGD step. For this reason, we scale $1 - \alpha_{\text{papa}}$ proportionally to the learning rate $\gamma$ (see Algorithm 1). This ensures that the trade-off between the gradients and PAPA remains the same when the learning rate changes.

## 2.4 Inference with the population

During training, we push the models toward the average. However, we still end up with multiple models when our goal is to have a single model. Knowing that we keep the models close to one another to make them amenable to averaging, the simplest solution is to return the average weights of the population at the end of training. A more intricate solution is to return the greedy model soup (Wortsman et al., 2022).

---

**Algorithm 2:** Averaging ($m = p$ for PAPA or PAPA-all, $m = 2$ for PAPA-2)

---

**Input:** $\Theta \leftarrow \{\theta_1, \cdots, \theta_p\}$, averaging from $m$ models;
**for** $i = 1 : p$ **do**
  $\{\theta'_1, \ldots, \theta'_m\} \leftarrow$ Randomly pick $m$ models without replacement
  $\hat{\theta}_i \leftarrow \frac{1}{m} \sum_{j=1}^{m} \theta'_j$;
**end**
$\hat{\Theta} \leftarrow \{\hat{\theta}_1, \ldots, \hat{\theta}_p\}$;
**return** $\hat{\Theta}$;

---

**Algorithm 3:** GreedySoup (Wortsman et al., 2022)

---

**Input:** $\Theta \leftarrow \{\theta_1, \cdots, \theta_p\}$, dataset $D$;
Sort $\Theta$ in decreasing order of accuracy on $D$;
$n \leftarrow 1$;
$\hat{\theta} \leftarrow \theta_1$;
**for** $i = 2 : p$ **do**
  $\hat{\theta}_{\text{new}} = \frac{n}{n+1}\hat{\theta} + \frac{1}{n+1}\theta_i$;
  **if** $accuracy(\hat{\theta}_{new}, D) \geq accuracy(\hat{\theta}, D)$ **then**
    $\hat{\theta} = \hat{\theta}_{\text{new}}$;
    $n \leftarrow n + 1$
  **end**
**end**
**return** $\hat{\theta}$;

---

**Model soups**   model soups average the weights of multiple pre-trained networks in some fashion to produce a single model (soup). Wortsman et al. (2022) propose two ways of constructing soups: 1) *average soup* consists in averaging the weights of all networks, which is precisely our population weight averaging (in Algorithm 2, the $m = p$ case); 2) *greedy soup* consists of sorting the networks from lower loss to higher loss, then, starting from the lowest-loss model, greedily adding the next model to the soup (through averaging the weights of the chosen models equally) if it reduces the loss of the soup (see Algorithm 3).

In PAPA, we construct average and greedy soups at the end of training and show that they both work similarly well in our specific setting to compress the population of trained models into a single network. For non-PAPA models, greedy soups work much better than average soups since there is no guarantee that the weights of all models are similar enough to be averaged without performance loss (in the worst case, greedy soups just take the network with the best validation accuracy).

# 3 Related work

**Concurrent work**   Diversify-aggregate-repeat training (DART) (Jain et al., 2023) is a concurrent work that was accepted to CVPR 2023 and appeared on arXiv one month prior to our first arXiv submission and for which we were made aware of afterwards.

Their approach, DART, uses almost the same setup as PAPA-all using average model soups with a similar motivation. There are a few minor differences: 1) they do not use REPAIR, 2) they wait until mid-training before averaging and average more rarely than us, 3) they apply one specific data augmentation per model (e.g., model 1 uses only mixup, model 2 uses only label smoothing, etc.), which limits how much models they can produce (while we randomly sample different hyperparameters from all data augmentations in each model), 4) they use Pad-Crop, AutoAugment, Cutout, and Cutmix as data augmentations, while we use Mixup, Label smoothing, CutMix, and Random Erasing, 5) they use a slightly different hyperparameters (600 epochs, weight decay of 5e-4, EMA by default).

As can be seen from this list, the differences are very minor. Thus, one can consider PAPA-all to be effectively the same algorithm as DART. The main difference between our work and DART is that we also propose PAPA (our best method across all PAPA variations) and PAPA-2, which are not considered in DART. Of note, DART also provides a nice theory for single-layer convolutional networks with some assumptions to show that using diverse data augmentations can reduce the convergence time to learn robust features while

averaging once in a while can increase the convergence time of noisy (non-meaningful) features (which is good since we ideally would prefer not learning the noise features). This theory also applies to PAPA-all and could potentially be extended to PAPA as future work.

In Section 4.5.2, we replicate DART and provide some comparisons between DART, PAPA-all, and PAPA. We observe that the differences between PAPA-all and DART are fairly minor and that the DART-specific design choices are not beneficial in terms of generalization.

**Federated learning and averaging over different data partitions**   Federated learning (Konečnỳ et al., 2016; McMahan et al., 2017) tackles the problem of learning when the data is scattered across multiple servers where exchanges of communication are rare. A separate model is trained in each server, and the goal is to combine those models into a single model in the central server. The main differences between classic federated learning and our setting are that 1) our models have access to the full dataset (with different data augmentations and regularizations) instead of partitions of the data, 2) we do not limit exchanges of communication, and 3) while federated learning seeks convergence of the central server model and generalization as close as possible to full-data training, we seek to generalize better than a single model trained on the full dataset.

Wang et al. (2020) use weight averaging in a federated learning setting. Su and Chen (2015); Li et al. (2022) also use weight averaging in partitioned-data setting, but with no limit to communication exchanges. Except for Li et al. (2022), a natural language processing (NLP) specific approach trained on different domains, these methods perform worse than a single model trained on the full training data since each model is trained only on a subset of the data. In contrast, we train each model on the full dataset and gain more performance than training a single model.

**Distributed Consensus Optimization**   Consensus optimization seeks to minimize a loss function while using different models per mini-batch and constraining their weights to be approximately equal (Xiao et al., 2007; Boyd et al., 2011; Shi et al., 2015). Similarly to federated learning, but contrary to us, the dataset is split across models. Consensus optimization algorithms are similar to PAPA because they push the weights toward the population average in order to reach a consensus (i.e., make all weights equal). However, reaching a consensus across all models is not the goal of PAPA. PAPA seeks to retain diversity between the models so that each model can discover different features to propagate through the population average.

**Genetic algorithms**   Genetic algorithms (GAs) (De Jong, 1975) seek to improve on a population of models by emulating evolution, a process involving: reproduction (creating new models by combining multiple models), random mutation (changing models randomly to explore the model space), and natural selection (retaining only the well-performing models). GAs have been used to train neural networks (Miller et al., 1989; Whitley et al., 1995; Such et al., 2017; Sehgal et al., 2019), but they are highly inefficient and computationally expensive.

Averaging weights during training can be interpreted as a form of reproduction. In fact, evolutionary stochastic gradient descent (ESGD) (Cui et al., 2018), a method that combines SGD with GAs, does reproduction by averaging the weights of two parents. Although PAPA-2 also averages two parents, it does not use any mutations nor natural selection, and we show that such components are actually harmful to generalization in Table 16. We also show that PAPA variants generalize better than ESGD in Appendix A.9.

**Averaging in optimization**   There is an extensive literature on averaging iterates (weights at each iteration) to speed up convergence with stochastic gradient descent (SGD) (Rakhlin et al., 2011; Lacoste-Julien et al., 2012; Shamir and Zhang, 2013; Jain et al., 2018). In the strongly convex setting, the convergence rate of SGD can be improved from $\mathcal{O}(\frac{\log T}{T})$ to $\mathcal{O}(\frac{1}{T})$ through averaging methods (Rakhlin et al., 2011; Lacoste-Julien et al., 2012). In the context of deep learning, similar averaging methods exist to improve generalization, such as exponential moving average (EMA) (Tarvainen and Valpola, 2017; Yazıcı et al., 2019) and stochastic weight averaging (SWA) (Izmailov et al., 2018) to name a few.

Contrary to PAPA, these techniques average the iterates from a single SGD trajectory rather than multiple models from different trajectories. Thus, these iterate averaging methods are orthogonal to PAPA and can

be combined with our approach. We demonstrate this in Appendix A.11 by showing that PAPA variants improve generalization when combined with SWA.

**Permutation-matching and mode connectivity** Freeman and Bruna (2017); Garipov et al. (2018); Draxler et al. (2018) showed that pre-trained networks could be connected through paths where the loss does not significantly decrease. However, for most networks, linear paths (interpolation between two or multiple networks) lead to a significant rise in loss. To alleviate this problem, various *permutation alignment* techniques have been devised. We tried some of these techniques (feature and weight matching from (Ainsworth et al., 2023) and Sinkhorn weight matching from (Peña et al., 2022)) but did not observe improvements using our algorithm.

Wortsman et al. (2022); Singh and Jaggi (2020); Matena and Raffel (2022) devise various strategies for combining multiple fine-tuned models to improve generalization. Instead, we show the more general statement that frequently pushing the models toward the average or occasionally averaging them during the whole length of training improves generalization (see Appendix A.12).

## 4 Experiments

For the experiments, we compare PAPA variants to baseline models trained independently with no averaging during training on two different tasks: image classification and satellite image segmentation. For image classification, we train models from scratch on CIFAR-10 (Krizhevsky et al., 2009), CIFAR-100 (Krizhevsky et al., 2009), and ImageNet (Deng et al., 2009); we also fine-tune pre-trained models on CIFAR-100. For image segmentation, we train models from scratch on ISPRS Vaihingen (Rottensteiner et al., 2012). On image classification, we only have access to train and test data; thereby, we remove 2% of the training data to use as evaluation data for the greedy soups.

We test different population sizes ($p \in [2, 10]$) with and without data augmentations and regularizations.

### 4.1 Choices of data augmentations and regularizations

Regarding the data augmentations and regularizations, we draw random hyperparameter choices for Mixup (Zhang et al., 2017; Tokozume et al., 2018), Label smoothing (Szegedy et al., 2016), CutMix (Yun et al., 2019), and Random Erasing (Zhong et al., 2020)).

Unless otherwise specified, we take a random draw from Mixup ($\alpha \in \{0, 0.5, 1.0\}$), Label smoothing ($\alpha \in \{0, 0.05, 0.10\}$), CutMix ($\lambda \in \{0, 0.5, 1.0\}$), and Random Erasing (probability of erasing a block of the image $\in \{0, 0.15, 0.35\}$)).

As an exception, for ImageNet, we use a random draw from Mixup ($\alpha \in \{0, 0.2\}$), Label smoothing ($\alpha \in \{0, 0.10\}$), CutMix ($\lambda \in \{0, 1.0\}$), and Random Erasing (probability of erasing a block of the image $\in \{0, 0.35\}$)).

### 4.2 Training hyperparameters

With PAPA, we apply the EMA every 10 SGD steps. For most experiments, we use $\alpha_{\text{papa}} = 0.99$ when training from scratch and $\alpha_{\text{papa}} = 0.999$ when fine-tuning. As exceptions, we use $\alpha_{\text{papa}} = 0.95$ for the ESGD experiments.

For training-from-scratch on CIFAR-10 and CIFAR-100, training is done over 300 epochs with a cosine learning rate (1e-1 to 1e-4) (Loshchilov and Hutter, 2016) using SGD with a weight decay of 1e-4. Batch size is 64 and REPAIR uses 5 forward-passes.

For training-from-scratch on ImageNet, training is done over 90 epochs with a cosine learning rate (1e-1 to 1e-4) (Loshchilov and Hutter, 2016) using SGD with a weight decay of 1e-4. Batch size is 256 and REPAIR is not used.

For fine-tuning, training is done over 150 epochs with a cosine learning rate (1e-4 to 1e-6) with and without restarts (every 25 epochs) (Loshchilov and Hutter, 2016) using AdamW (Kingma and Ba, 2014; Loshchilov and Hutter, 2017) with a weight decay of 1e-4. Since we use a pre-trained model, the final layer is changed to fit the correct number of classes and re-initialized with random weights. As recommended by Kumar et al. (2022), we freeze all layers except the final layer for the first 6 epochs of training (4% of the training time). Batch size is 64 and REPAIR uses 5 forward-passes.

### 4.3 Presentation

For simplicity, only the results with the largest population size and with data augmentations are shown in the main tables, while the rest is left in the Appendix. We report the test accuracies of the logit-average ensemble (Ensemble) (Tassi et al., 2022) for all models. As mentioned before, average soups perform poorly for baseline models, while average soups tend to perform better than greedy soups for PAPA variants. Thus, for parsimony in the main result table, we only show the average soup (AvgSoup) for PAPA variants and greedy soup (GreedySoup) for baseline models. Detailed results with both soups are found in the Appendix.

To increase the readability of the lengthy tables, we highlight the metrics of the best ensemble in **green** and the best model soup in **blue** for each setting across all approaches and population sizes. When results are written as "x (y)", x is the mean accuracy, and y is the standard deviation across 3 independent runs (with seeds 1, 2, 3); otherwise, it is the accuracy of a single run with seed=1.

For the models trained from scratch on CIFAR-10 and CIFAR-100, we tried to improve the model soups of the baseline approach by using permutation alignment through feature matching (Ainsworth et al., 2023) on the full training data; however, permutation-alignment is only implemented on specific architectures due to its complexity of implementation, and since it did not improve greedy soups while average soups still performed poorly (1-45% accuracy while regular models had 73-81 % accuracy), we did not use feature alignment for soups in the other settings. Meanwhile, REPAIR was helpful to baseline soups, and its overhead cost was negligible, so we used it to improve the baseline model soups. See Appendix A.4 for more details on the permutation alignment.

For the full details on the datasets, neural network architectures, training hyperparameters, and data augmentations and regularizations, please refer to Appendix A.1.

### 4.4 Main experiments

We show our main results (with varying augmentations at the largest population size we trained) in Table 1 and leave the detailed results (with and without augmentation, with various population sizes) in Appendix A.7 and those on Vaihingen in Appendix A.8.

Table 1: Test accuracy from ensembles and soups with varying data augmentations and regularizations

| Dataset / architecture | Baseline | | PAPA | | PAPA-all | | PAPA-2 | |
|---|---|---|---|---|---|---|---|---|
| | Ensemble | GreedySoup[1] | Ensemble | AvgSoup | Ensemble | AvgSoup | Ensemble | AvgSoup |
| | $p$ models | 1 model | $p$ models | 1 model | $p$ models | 1 model | $p$ models | 1 model |
| **CIFAR-10** ($n_{epochs} = 300$, $p = 10$) | | | | | | | | |
| VGG-11 | **95.2** (0.1) | 94.0 (0.1) | 94.9 (0.1) | **94.8** (0.0) | 94.1 (0.2) | 94.1 (0.2) | 94.5 (0.1) | 94.4 (0.1) |
| ResNet-18 | **97.5** (0.0) | 96.8 (0.2) | 97.4 (0.1) | **97.4** (0.1) | 97.3 (0.1) | 97.3 (0.1) | 97.1 (0.0) | 97.1 (0.1) |
| **CIFAR-100** ($n_{epochs} = 300$, $p = 10$) | | | | | | | | |
| VGG-16 | **82.2** (0.1) | 77.8 (0.1) | 79.6 (0.4) | **79.4** (0.3) | 79.0 (0.4) | 78.9 (0.4) | 79.0 (0.3) | 78.9 (0.3) |
| ResNet-18 | **84.3** (0.3) | 80.2 (0.6) | 82.2 (0.1) | **82.1** (0.2) | 81.8 (0.0) | 81.8 (0.0) | 81.3 (0.3) | 81.2 (0.3) |
| **Imagenet** ($n_{epochs} = 90$, $p = 3$) | | | | | | | | |
| ResNet-50 | **78.7** | 76.8 | 78.4 | **78.4** | 77.7 | 77.7 | 77.8 | 77.8 |
| **Fine-tuning on CIFAR-100** ($n_{epochs} = 50$, $p = 2, 4, 5$ respectively for EfficientNetV2, EVA-02, ConViT) | | | | | | | | |
| EffNetV2-S | **91.7** (0.3) | 91.3 (0.4) | 91.6 (0.3) | **91.4** (0.5) | 91.4 (0.4) | 91.1 (0.4) | 91.3 (0.6) | 91.3 (0.6) |
| EVA-02-Ti | 90.6 (0.1) | 90.4 (0.1) | **90.7** (0.3) | 90.6 (0.2) | **90.7** (0.6) | **90.7** (0.5) | 90.5 (0.3) | 90.4 (0.3) |
| ConViT-Ti | **88.8** (0.2) | 87.9 (0.2) | 88.6 (0.2) | **88.4** (0.2) | 88.2 (0.2) | 88.1 (0.1) | 88.2 (0.2) | 88.2 (0.3) |

---

[1]Note that when training from scratch (the non-fine-tuning results), the greedy soup is just the best model (based on validation accuracy) since the models are not amenable to averaging. See Section A.13 for details.

In the table, we show the different approaches in the columns (Baseline, PAPA, PAPA-all, and PAPA-2) using ensembles of $p$ models or model soups (a single condensed model). In the rows, we show the different architectures for different datasets.

From the experiments, we observe the following elements: PAPA tends to perform better than other PAPA variants, while PAPA-all and PAPA-2 perform better than baseline models. Baseline ensembles have higher accuracy than PAPA variants. For PAPA variants, average soups tend to generalize better than greedy soups. Baseline greedy groups only used one model when pre-training but multiple models when fine-tuning (See Appendix A.13); this shows that pre-trained models cannot be merged without a drop in performance when not using PAPA variants. Increasing $p$ tends to improve the test accuracy of PAPA variants. Varying data augmentations and regularizations generally leads to the highest performance for ensembles and soups. Thus, optimal performance is found with an ensemble of models; however, if one seeks to maximize performance for single networks, PAPA is the way to go.

**ImageNet**   PAPA obtains an accuracy of 78.36% on ImageNet. Previous results with similar accuracy were obtained using large batch sizes (1024 for 300 epochs, 2048 for 100 epochs) or long training (384 for 600 epochs) (Wightman et al., 2021). Meanwhile, we use only a batch size of 256 for 90 epochs with 3 networks (equivalent in compute to 270 epochs on a single GPU). We also observe that PAPA is better than its variants PAPA-all and PAPA-2 by a safe margin of at least 0.6%.

**Fine-tuning**   Greedy model soups have been shown to perform best in the context of fine-tuning from a pre-training model (Wortsman et al., 2022). Nevertheless, we see an improvement in generalization from using PAPA (with a very small $\alpha_{\text{papa}} = 0.9995$), which shows that leveraging the population average during fine-tuning is still beneficial.

## 4.5   Additional experiments

### 4.5.1   Comparing PAPA variants to baseline models trained for $p$ times more epochs

Previously we compared PAPA variants to baseline models trained on the same amount of epochs. However, one may argue that PAPA approaches have used $p$ times more training samples due to the updates with the population average of the weights. Furthermore, suppose one has access to a single GPU. In that case, one may wonder whether they should train a single model for $p$ times more epochs or use PAPA variants with $p$ networks since having a single GPU means they cannot leverage the parallelization benefits of our method.

Here we compare the accuracy of independent (baseline) models trained for 3000 epochs to PAPA variants trained with $p = 10$ networks for 300 epochs. For the baseline models, we randomly select from the set of augmentations and regularizations in each mini-batch to replicate the effect of training on all these data variations.

Results are shown in Table 2. We find that all PAPA approaches (PAPA-all, PAPA, PAPA-2) generalize better than baseline models trained 10 times longer, except in the case of ResNet-18 with no data augmentation, where only PAPA-all generalize better than baseline models.

This demonstrates that PAPA variants can bring more performance gain than training models for more epochs. This suggests that our approach could provide a better and more efficient way of training large models on large data by parallelizing the training length over multiple networks trained with PAPA variants for less time.

### 4.5.2   Comparing PAPA to DART

As mentioned in Section 3, DART (Jain et al., 2023) is a concurrent work, which is very similar to PAPA-all. The main differences are the different data augmentations, lack of REPAIR, start of the averaging at mid-training, and less frequent averaging. In this section, we try to replicate DART by using PAPA-all with the DART-specific design choices and training hyperparameters. We also provide ablation to determine the usefulness of each of the DART-specific design choices.

Table 2: Training independent models for $300 \times 10$ epochs versus training $p = 10$ PAPA models for 300 epochs on CIFAR-100

| Baseline | PAPA | PAPA-all | PAPA-2 |
|---|---|---|---|
| Mean | AvgSoup | AvgSoup | AvgSoup |
| **VGG-16: No data augmentations or regularization** | | | |
| 74.15 (0.1) | **76.04** | 75.13 | 75.10 |
| **VGG-16:With random data augmentations** | | | |
| 77.44 (0.1) | **79.36** (0.3) | 78.89 (0.4) | 78.91 (0.3) |
| **ResNet-18: No data augmentations or regularization** | | | |
| 78.23 (0.6) | 78.11 | **78.59** | 77.90 |
| **ResNet-18:With random data augmentations** | | | |
| 79.88 (0.5) | **82.06** (0.2) | 81.77 (0.0) | 81.23 (0.3) |

In Jain et al. (2023), they ran experiments on CIFAR-10/CIFAR-100 with a ResNet-18 network. Sadly, they did not release the code for these experiments, and some hyperparameters are unclear. They use an exponential moving average (EMA) of the weights but do not specify the EMA rate nor how they merge the EMA networks. Furthermore, they use three networks, each with one specific data augmentation: AutoAugment, Cutout, and Cutmix; however, they do not specify the hyperparameters of these transformations.

We try to replicate those experiments; however, keep in mind that we likely do not have the correct hyperparameters for the EMA and the data augmentations.

We train ResNet-18 models on CIFAR-10 and CIFAR-100 with a population of $p = 3$ networks. The first network uses AutoAugment from (Wightman et al., 2021) with $m = 4$, $n = 1$, $p = 1.0$, $mstd = 1.0$ (i.e., select one transformation per image with magnitude 4). The second network uses Cutmix with $\lambda = 0.5$. The third network uses Cutout (DeVries and Taylor, 2017) with one hole of length 8. Following their protocol, the training is done over 600 epochs with a cosine learning rate (0.1) with weight decay (5e-4).

We try PAPA, PAPA-all, and DART. We also provide some ablation to determine the usefulness of the DART-specific design choices, which are not in PAPA-all. Table 3 shows the results. We see that within our replication, DART generalizes a little bit less than PAPA-all. The best generalization is obtained with PAPA-all without REPAIR for CIFAR-10 and PAPA-all (with REPAIR) for CIFAR-100. This shows that the design choices to start averaging at mid-training and average less frequently are not particularly useful. In fact, Wightman et al. (2021) mention in their own ablation that the results are relatively insensitive to these design choices.

The results of our replication are similar to those reported in Jain et al. (2023), except for slightly worse performance on CIFAR-100. Keep in mind that we do not have the exact training details and cannot ensure the exact same setup as them. Note that the results we obtain here are different from those of our main results (Table 1) because we use 600 epochs instead of 300 and completely different data augmentations.

Overall, the small design differences between DART and PAPA-all are not that important, and ultimately, the two methods are nearly the same.

Table 3: PAPA and PAPA-all average soups vs DART with ResNet-18 on a population of $p = 3$ networks

| Method | CIFAR-10 | CIFAR-100 |
|---|---|---|
| Original results by Jain et al. (2023) | | |
| ERM+EMA (Mixed - MT) | 97.08 (0.05) | 82.25 (0.29) |
| DART | **97.14** (0.08) | **82.89** (0.07) |
| PAPA, PAPA-all, and DART replication | | |
| PAPA-all | 97.10 (0.15) | **82.59** (0.24) |
| PAPA-all no-repair | **97.20** (0.06) | 82.29 (0.42) |
| PAPA-all no-repair, starts at 300 epochs | 97.12 (0.09) | 82.26 (0.18) |
| DART: PAPA-all no-repair, starts at 300 epochs, average every 40/50 epochs | 97.12 (0.13) | 82.30 (0.19) |
| PAPA | 97.15 (0.09) | 82.45 (0.16) |

### 4.5.3 Stochastic weight averaging (SWA)

We run experiments with stochastic weight averaging (SWA) on CIFAR-100 with $p = 5$ in Appendix A.11. We find that using SWA increases the mean accuracy of baseline networks from 79.35% to 80.77% and that further adding PAPA-all brings the mean accuracy to 81.88%. This demonstrates that SWA and PAPA variants are orthogonal methods that work for different reasons and complement one another. Thus, for optimal performance, we suggest combining PAPA with SWA.

### 4.5.4 Evolutionary stochastic gradient descent (ESGD)

We compare PAPA variants to evolutionary stochastic gradient descent (ESGD) (Cui et al., 2018), a method that leverage a population of networks through a complicated genetic algorithm, on the same CIFAR-10 setting explored by (Cui et al., 2018). We show that PAPA-all and PAPA obtain similar or better results with only 5 networks instead of 128. Results are shown in Appendix A.9.

### 4.5.5 Logistic regression (single-layer model)

We test a logistic regression setting to see how PAPA fares on single-layer models. Results are shown in Appendix A.10. These results demonstrate that model soups can already work well without alignment in the single-layer setting and that PAPA benefits are specific to deep neural networks.

### 4.5.6 Ablations

We provide an extremely detailed ablation study in Appendix A.12 showing the drop in performance from removing/changing parts of the algorithm. We also investigate the effect of pushing toward the average or replacing by the average solely during certain periods of training (e.g., beginning, middle, end) instead of during the entire length of training. We observe that mean accuracy is highest when averaging during the whole span of training time (as done in our method).

When $\alpha_{\mathrm{papa}} = 0$, we generally see that averaging at different frequencies, using the same initialization for every model, removing REPAIR, using feature-matching permutation alignment (Ainsworth et al., 2023), and adding elements of Genetic Algorithms decrease performance; the only exception is with PAPA-all where averaging every 10 epochs (instead of 5) performs better.

When $\alpha_{\mathrm{papa}} > 0$, we show that applying an EMA of $\alpha_{\mathrm{papa}} = 0.99$ every 10 iterations work best.

### 4.6 Visualizations of the accuracy and its change after averaging

We also provide visualizations of how the accuracy changes over time in Figure 2. We observe that 1) test accuracy is almost always higher with PAPA variants than with baseline models, 2) REPAIR prevents significant loss in accuracy after averaging with PAPA-all in the first few epochs, but has little effect afterwards, and 3) test accuracy is massively boosted immediately after averaging with PAPA-all, then it drops significantly from the first few iterations of training, and then it slowly increases again until the next averaging.

### 4.7 Why does averaging the weights of a population help so much?

We hypothesize that averaging is beneficial because we effectively combine features from one network with the features of another, allowing the averaged model to learn unrealized features discovered by other networks. We provide evidence for this hypothesis by showing that 1) PAPA models (and variants) share most of their features at the end of training (showing feature mixing), and 2) performance immediately improves after averaging (showing a benefit from mixing features).

To show feature mixing, we align the networks of a population (through feature matching permutation alignment (Ainsworth et al., 2023) and REPAIR (Jordan et al., 2023)) and demonstrate that PAPA variants have higher cosine similarities (See Appendix A.5 for the details): 88–99% between networks of the PAPA variants, and 31–63% for independent models. To show the benefit of feature mixing, we look at the change

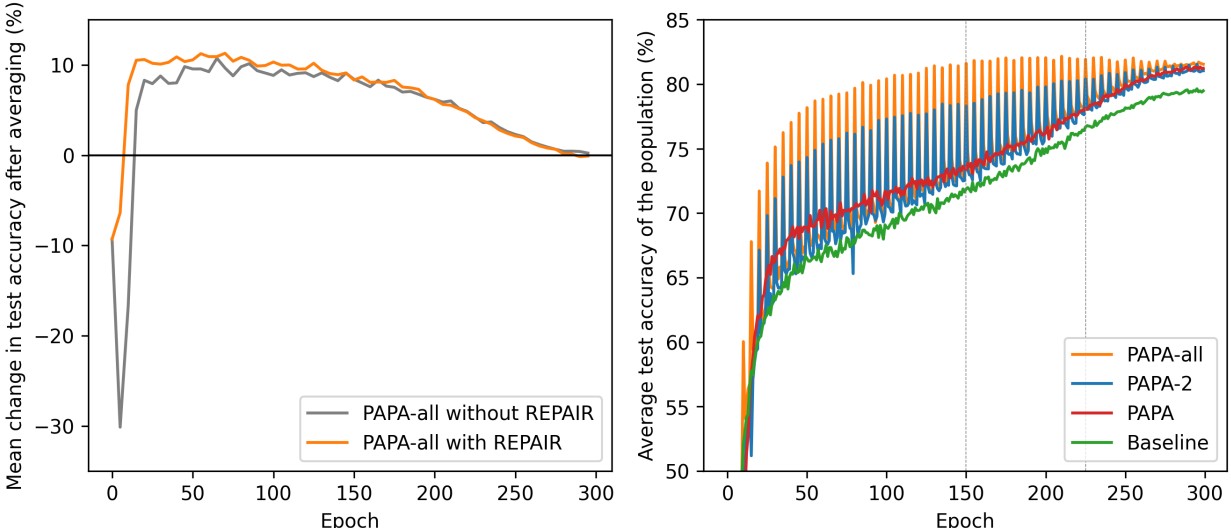

Figure 2: Accuracy (and its change after averaging) at each epoch with PAPA variants on CIFAR-100.

in accuracy after averaging with PAPA-all and PAPA-2 in Figure 2, and we observe a massive increase in accuracy (around 10% increase) immediately after averaging. These results demonstrate feature mixing and that mixing these features significantly boosts performance.

## 5 Discussion

During pre-training, greedy model soups only used a single model and performed worse than PAPA variants. During fine-tuning, greedy model soups used multiple models. In the case of training a single layer, our experiments showed that PAPA performed worse than greedy soups. The conclusion from these experiments is the following:

1. During pre-training, PAPA ensures that the models stay similar enough so that model averaging provides equal or greater generalization. Meanwhile, model soups will only choose the best network because the models are not amenable to averaging, and doing so would lead to a massive drop in generalization.

2. During fine-tuning from a pre-trained model, models are amenable to averaging, and thus, PAPA is unnecessary over model soups. We still found small benefits of PAPA over model soups in the case of fine-tuning all parameters. However, this is because we fine-tuned all parameters. When training a single layer (as is the case for linear probing), PAPA is unnecessary because averaging the weights of a single linear layer already works since the permutations are perfectly aligned.

## 6 Conclusion

We present an algorithm called PopulAtion Parameter Averaging (PAPA), which trains a population of $p$ models and improves overall performance by pushing the weights toward the population average. We also propose PAPA variants (PAPA-all, and PAPA-2) that rarely replace the weights with the population average. In practice, we find that all methods increase generalization, but PAPA tends to perform better than PAPA-all and PAPA-2. PAPA provides a simple way of leveraging a population of networks to improve performance. Our method is easy to implement and use. Our experiments use a single GPU, demonstrating that PAPA is worthwhile with small compute and could be scaled much further.

In practice, PAPA performed better than training a single model for $p$ times more epochs. Thus, PAPA could provide a better and more efficient way of training large models on extensive data by parallelizing the training length over multiple PAPA networks trained for less time.

**Limitations** PAPA is more expensive than training a single network. Also, ensembles perform better than PAPA, though at a high inference costs. When training multiple networks with no computational or memory constraint at inference, one is better off using ensembles; however, when such constraints exist, or one needs a single model, one is better off using PAPA soups. The benefit of PAPA is significant for pre-training but small for fine-tuning because the weights stay relatively well aligned.

**Future work** We only explored equal-weighted averaging, but different weightings may help emphasize more generalizable features. To parallelize PAPA, one could test ideas such as asynchronous updates. Theory from consensus optimization may help prove the generalization benefits of PAPA.

## Acknowledgments

This research was enabled in part by compute resources provided by Mila (`mila.quebec`), Calcul Québec (`calculquebec.ca`), the Digital Research Alliance of Canada (`alliancecan.ca`), and by support from the Canada CIFAR AI Chair Program. KF is supported by NSERC Discovery grant (RGPIN-2019-06512) and a Samsung grant. Simon Lacoste-Julien is a CIFAR Associate Fellow in the Learning Machines & Brains program.

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

# A  Appendix

## A.1  Datasets and training details

### A.1.1  Datasets

Table 4: Datasets

| Dataset | Image size (original) | Image size (after resizing) | # of classes | # of images | | | License |
|---|---|---|---|---|---|---|---|
| | | | | train | test | valid | |
| CIFAR-10 | 32x32 | 32x32 | 10 | 50K | 10K | N/A | N/A |
| CIFAR-100 | 32x32 | 32x32 | 100 | 50K | 10K | N/A | N/A |
| ImageNet | varies | 224x224 | 1000 | 1.28M | 50K | N/A | N/A |
| Vaihingen | 2494x2064 | 256x256 | 6 | 11 | 17 | 5 | N/A |

Vaihingen has a validation set, but there is no validation set for the other datasets, in which case, we keep 2% of the training as held-out data to compute the validation accuracy for the greedy model soups.

### A.1.2  GPUs

For all experiments, we use a single GPU: A-100 40Gb (for Imagenet) or V-100 16Gb (for all other experiments).

### A.1.3  Vaihingen

For the Vaihingen dataset (Rottensteiner et al., 2012), we follow the training procedure and PyTorch implementation from (Audebert et al., 2017). We use a UNet (Ronneberger et al., 2015) and the train, validation, and test splits from (Fatras et al., 2021). We use 11 tiles for training, 5 tiles for validation, and the remaining 17 tiles for testing our model. Furthermore, we only consider the RGB components of the Vaihingen dataset (Rottensteiner et al., 2012). We train our models with augmented (flip and mirror) $256 \times 256$ patches from training data. The size of an epoch is set to 10000 images, and we train our model for 50 epochs. We use an initial learning rate of 0.01 and divide it by ten at 50% and 90% of training. For validation and test datasets, we directly apply our model over the original tiles.

### A.1.4  Network architectures

We use VGG-16 (Simonyan and Zisserman, 2014), ResNet-18, ResNet-20, ResNet-50 (He et al., 2016), EfficientNetV2 (Tan and Le, 2021), EVA-02 (Fang et al., 2023), and ConViT (d'Ascoli et al., 2021), and UNet (Ronneberger et al., 2015).

## A.2  Techniques used or considered in PAPA-all and PAPA-2

**REPAIR**  Jordan et al. (2023) observed that interpolating between two networks can lead to a collapse of the variance of features after interpolation. REnormalizing Permuted Activations for Interpolation Repair (REPAIR) is a technique to mitigate the variance collapse through rescaling the preactivations (i.e., the convolution/linear layers arising before any activation function). The rescaling is done by reweighting the bias and slope of the convolution (or linear) layers so that the preactivation features are such that:

$$\mathbb{E}[X_\alpha] = (1 - \alpha) \cdot \mathbb{E}[X_1] + \alpha \cdot \mathbb{E}[X_2], \tag{2}$$
$$\mathrm{std}(X_\alpha) = (1 - \alpha) \cdot \mathrm{std}(X_1) + \alpha \cdot \mathrm{std}(X_2). \tag{3}$$

where $X_\alpha$, $X_1$, $X_2$ are the features of the interpolated, first, and second networks. This method enforces the first and second moments of the features of the interpolated network to be equal to the interpolation of these moments from the two original networks. REPAIR requires a few forward passes of the training data to calculate approximations of the expectations and standard deviations.

We found REPAIR beneficial in reducing the performance loss from averaging, especially early in training, and obtaining a slight but consistent improvement in performance. However, given the compute cost of

REPAIR, we only used it on PAPA-all and PAPA-2. Note that we generalized REPAIR from two networks to $p$ networks on varying data augmentations and modified the algorithm to ensure that every moment of the features uses the same data (except for data augmentations) to minimize the variance of the estimated feature moments. The modified algorithm is described in Appendix A.3.

**Permutation alignment**   Although we use it on its own, REPAIR was initially a method to improve on the recent permutation alignments techniques (Ainsworth et al., 2023; Entezari et al., 2022; Peña et al., 2022), a set of approaches that have shown great promise in improving the accuracy of models after interpolation. These methods consist in permuting the layers of the first neural network to make the features or the weights (depending on the technique) closest to those of the second network (before interpolating the two networks). Permuting the layers is done in such a way that the output of the neural network is the same, but the difference is that when interpolating between the two networks, their features/weights are now well aligned.

Since we average the weights of many networks, this direction seemed promising. However, we found no benefits of permutation alignment before averaging; we attribute this to the fact that the models are recombined often enough to keep them similar permutation-wise so that averaging them does not lead to accuracy loss (in fact, we generally see a big improvement in accuracy from averaging).

## A.3   REPAIR

We describe our slightly modified REPAIR algorithm, which we used in Algorithm 4. The main changes over the original algorithm from Jordan et al. (2023) are that we generalized it from 2 networks to $p$ networks, used different data augmentations per network, and reused the same seed to ensure all networks go through the same data (except for data augmentations) to minimize the variance of the estimated feature moments. Note that although quite complicated, the algorithm is easy to use and can be directly applied to most network architectures because of the batch-norm insertion trick.

---

**Algorithm 4:** REPAIR (Jordan et al., 2023)

---

**Input:** $\Theta \leftarrow \{\theta_1, \cdots, \theta_p\}$, train (or evaluation; but not test) dataset $D$, set of data augmentations and regularizations $\{\pi_j\}_{j=1}^p$, $j$ is the current network being repaired, SEED=666, $k = 5$ iterations (forward passes), $w = (w_1, \cdots, w_p)$ weights for the averaging (default is $w_i = \frac{1}{p} \forall i$).

// Definitions

Let $\alpha$, $\beta$, $\mu$, $\sigma$ be the bias, weight, running mean, and running standard deviations of a batch-norm layer

Let $a$, $b$ be the bias, and weight of a pre-activation (linear or convolution) layer

// Set temporary batch norms to calculate features statistics

$\theta' = \theta_j$;

Add a temporary batch-norm layer after all pre-activation layers of $\theta'$ and include the batch-norm in the forward pass;

**for** $i = 1 : p$ **do**

    Add a temporary batch-norm layer after all pre-activation layers of $\theta_i$, but ignore the batch-norm in the forward pass;

    // Rebuild the batch-norm statistics from scratch

    ResetBatchNorm($D$, $\pi_i$, $\theta_i$, SEED, $k$);

**end**

// Calculate weighted running mean and standard deviations

$\mu = [0, \cdots, 0]$;

$\sigma = [0, \cdots, 0]$;

**for** $i = 1 : p$ **do**

    **for** *temporary batch-norm layer $l$ of $\theta_i$* **do**

        Extract $\mu$ and $\sigma$ from $l$;

        $\mu(l) = \mu(l) + w_i \mu'$;

        $\sigma(l) = \sigma(l) + w_i \sigma'$;

    **end**

**end**

// Replace preactivation weight and bias by weighted running mean and standard deviation

**for** *temporary batch-norm layer $l$ of $\theta'$* **do**

    Set $\alpha = \mu(l)$ and $\beta = \sigma(l)$ in layer $l$;

**end**

// Reset the running mean and standard deviations

ResetBatchNorm($D$, $\pi_j$, $\theta'$, SEED, $k$)

// Fuse the rescaling coefficients into the preactivation

**for** *(preactivation layer $l_1$ and subsequent temporary batch-norm $l_2$) of $\theta'$* **do**

    Extract $\alpha$, $\beta$, $\mu$ and $\sigma$ from $l_2$;

    Extract $a$, $b$ from $l_1$;

    Set $b = \frac{\beta}{\sigma} b$ in layer $l_1$;

    Set $a = \alpha + \frac{\beta}{\sigma}(a - \mu)$ in layer $l_1$;

    Remove the temporary batch-norm layer $l_2$

**end**

**return** $\theta'$;

---

---

**Algorithm 5:** ResetBatchNorm

---

**Input:** train (or evaluation; but not test) dataset $D$, $\theta$, $\pi$, SEED $= 666$, $k = 5$.

`// Reset the batch-norm statistics to a running mean of 0 and standard-deviation of 1`
Reset the running stats of all batch-norm layers in $\theta$;
`// Rebuild the batch-norm statistics from scratch`
Set the random seed to SEED;
**for** $i = 1 : k$ **do**
   | Sample data from $D$
   | Regularize data using $\pi$
   | Forward pass through $\theta$;
**end**

---

### A.4 Notes on the permutation alignments

As previous mentioned in Section 4, in the model soups of some non-PAPA models (the VGG and Resnet-18 models on CIFAR-10 and CIFAR-100) and some ablation experiments, we use permutation alignment methods to align the different networks to obtain better averaging or model soups. We use the feature-matching method by Ainsworth et al. (2023) for the permutation alignment of the paper. We want to acknowledge that we also did preliminary experiments with weight matching (Ainsworth et al., 2023) and the Sinkhorn implicit differentiation (Peña et al., 2022). However, we obtained noticeably worse alignments with these methods than the feature-matching one. Thus, we stuck to the feature-matching method in this paper.

Below, we discuss some of the challenges with aligning multiple networks and how we addressed these problems.

Although permutation alignment is well-defined for aligning two networks, the problem of aligning multiple networks is ill-defined. Which criterion to optimize needs to be clarified and depends on how you want to use these networks (e.g., simple averaging versus making model soups). One option is to simultaneously learn all permutations and minimize the average distance in weight space between any pair of networks. However, this approach is memory-expensive and may not be optimal if the goal is to make, for example, greedy soups. Below, we discuss how we chose to align multiple networks.

For the permutation alignment with PAPA-all (which we only use in the ablation of Appendix A.12), we merely align every network to the first network of the population; this may not be perfect, but given the poor results we obtained, it did not feel worthwhile to pursue other ways of aligning the networks.

For aligning the baseline models to make (average or greedy) model soups, we take advantage of how the greedy soup is built to align the models better. We sort the models in decreasing order of train/validation accuracy (as done in greedy soups). Then, before adding the next considered model into the soup, we permute-align this network with respect to the current soup. This ensures that the considered model is well aligned with the soup before being added.

**Extra notes about the code** The code (from Ainsworth et al. (2023)) we use for permutation alignment is very specific to VGG and ResNet18 architectures. Generalizing this code to handle arbitrary networks, especially those with many residual layers and submodules, is highly non-trivial. In contrast, the code of REPAIR (Jordan et al., 2023) generalizes to nearly every modern neural network architecture.

### A.5 Similarity between features learned from PAPA variants

To test the hypothesis that features are mixed while using PAPA variants, we analyze the cosine similarity of the networks' features at the end of training after doing feature-matching (Ainsworth et al., 2023) on full training data and REPAIR (Jordan et al., 2023) with five iterations. The feature-matching and REPAIR used with the Baseline approach are needed to align the features between the networks to ensure that we don't compare apples to oranges; this is similar to what is done in Li et al. (2015) to analyze how much

features are shared between networks. We used the setting with CIFAR-100 data and ResNet-18 with $p = 5$, which is the same setting as in the Ablation of Appendix A.12.

We observe that independently trained models have cosine similarities varying between 30.87% and 63.13%, while models trained with PAPA variants, which were not averaged for the last five epochs, have cosine similarities between 87.79% and 99.80%. This shows that averaging indeed ensures good mixing of the features.

Interestingly, networks of PAPA have slightly lower similarity than those of PAPA variants; this can be explained by the slow push toward the average. This also explains why PAPA tends to generalize better than other PAPA variants since it enables good averaging while retaining more diversity in the models.

Table 5: Cosine similarity at different hidden layers for CIFAR-100 with ResNet-18

| Layer | 1 | 2 | 3 | 4 |
|---|---|---|---|---|
| **Baseline** | | | | |
| Between two networks | 48.07 | 48.78 | 36.76 | 30.71 |
| Between one network and AvgSoup | 63.13 | 55.13 | 41.80 | 50.15 |
| **PAPA** | | | | |
| Between two networks | 98.49 | 96.88 | 94.04 | 87.79 |
| Between one network and AvgSoup | 99.31 | 98.68 | 97.61 | 94.29 |
| **PAPA-all** | | | | |
| Between two networks | 99.61 | 99.12 | 98.73 | 97.12 |
| Between one network and AvgSoup | 99.80 | 99.60 | 99.41 | 98.39 |
| **PAPA-2** | | | | |
| Between two networks | 99.41 | 98.58 | 97.81 | 95.56 |
| Between one network and AvgSoup | 99.76 | 99.37 | 99.07 | 97.61 |

## A.6 Why adding aspects of Genetic Algorithms (GAs) to PAPA variants is harmful

As discussed in Section 3 and shown in Appendix A.12, incorporating elements of GAs worsens the performance gain obtained from averaging. There is a clear explanation for why incorporating aspects of Genetic algorithms (GAs) with PAPA variants is detrimental.

First, adding random white noise to weights (mutations) is a very noisy way of exploring the data; thus, it is unsurprising that adding mutations without natural selection is harmful. We produce diversity in a more reliable way through random data ordering, augmentations, and regularizations.

Second, natural selection (an elaborate way to say: selecting which subset of models will be used for reproduction/averaging, which includes greedy soups) prevents us from using the entire population. Remember that the key to averaging is to bring the weights close enough to the average or average often enough so that the weights are not too dissimilar (which leads to poor averaging). During one epoch, assume that model 1 is not selected, or maybe selected only to be averaged with model 2 but not the rest of the population; the end result is that you will end up with specific clusters of models that are averaged together (or a few single models never averaged) because the longer two models are not averaged together, the more dissimilar these models will become. When using any model selection in our ablation (greedy soups included), we see the result of this behavior: the average soup performs badly at the end of training. This suggests that good performance with GAs would require a substantial population size to ensure that the largest cluster of similar (easy-to-average) models contains enough models at the end of training.

## A.7 Detailed results

We show the detailed results on CIFAR-10, CIFAR-100, and Imagenet below.

Table 6: Test accuracy (Ensemble, Average and Greedy soups) on CIFAR-100 with VGG-16

| | Baseline | | | PAPA | | | PAPA-all | | | PAPA-2 | | |
|---|---|---|---|---|---|---|---|---|---|---|---|---|
| | | Soups | | | Soups | | | Soups | | | Soups | |
| | Ens | Avg | Greed | Ens | Avg | Greed | Ens | Avg | Greed | Ens | Avg | Greed |
| # of models | $p$ | 1 | 1 | $p$ | 1 | 1 | $p$ | 1 | 1 | $p$ | 1 | 1 |
| **No data augmentation or regularization** | | | | | | | | | | | | |
| $p = 3$ | **77.72** | 41.18 | 74.00 | 74.93 | 74.90 | 74.84 | 74.41 | 74.43 | 74.39 | 75.28 | **75.29** | **75.16** |
| $p = 5$ | **78.63** | 4.58 | 74.20 | 75.97 | **76.01** | **76.07** | 74.99 | 75.00 | 75.06 | 75.25 | 75.26 | 75.29 |
| $p = 10$ | **79.24** | 1.00 | 73.77 | 76.03 | **76.04** | **76.17** | 75.12 | 75.13 | 75.06 | 75.11 | 75.10 | 75.21 |
| **With random data augmentations** | | | | | | | | | | | | |
| $p = 3$ | **80.42** | 29.41 | 77.65 | 78.9 | 78.72 | 78.54 | 78.89 | **78.74** | **78.66** | 78.56 | 78.64 | 78.62 |
| $p = 5$ | **80.98** | 3.24 | 77.26 | 79.23 | **79.06** | **78.72** | 78.71 | 78.68 | 78.28 | 78.52 | 78.50 | 78.09 |
| $p = 10$ | **82.03** | 0.97 | 77.72 | 79.88 | **79.65** | **79.08** | 78.71 | 78.60 | 78.50 | 79.22 | 79.14 | 78.79 |

Table 7: Test accuracy (Ensemble, Average and Greedy soups) on CIFAR-100 with ResNet-18

| | Baseline | | | PAPA | | | PAPA-all | | | PAPA-2 | | |
|---|---|---|---|---|---|---|---|---|---|---|---|---|
| | | Soups | | | Soups | | | Soups | | | Soups | |
| | Ens | Avg | Greed | Ens | Avg | Greed | Ens | Avg | Greed | Ens | Avg | Greed |
| # of models | $p$ | 1 | 1 | $p$ | 1 | 1 | $p$ | 1 | 1 | $p$ | 1 | 1 |
| **No data augmentation or regularization** | | | | | | | | | | | | |
| $p = 3$ | **79.36** | 27.99 | 76.49 | 77.87 | **77.89** | **77.86** | 77.48 | 77.48 | 77.50 | 77.18 | 77.19 | 77.38 |
| $p = 5$ | **79.96** | 7.10 | 77.02 | 77.67 | 77.69 | 77.71 | 77.99 | **78.01** | **78.02** | 77.71 | 77.71 | 77.8 |
| $p = 10$ | **80.54** | 1.50 | 76.78 | 78.15 | 78.11 | 78.06 | 78.56 | **78.59** | **78.40** | 77.85 | 77.90 | 77.90 |
| **With random data augmentations** | | | | | | | | | | | | |
| $p = 3$ | **82.89** | 24.69 | 80.86 | 81.99 | 81.73 | 81.46 | 81.24 | 81.29 | 81.30 | 81.77 | **81.74** | **81.74** |
| $p = 5$ | **83.08** | 8.68 | 80.69 | 81.67 | 81.52 | 80.88 | 81.19 | 81.03 | 81.50 | 81.62 | **81.59** | **81.68** |
| $p = 10$ | **84.38** | 1.00 | 79.94 | 82.28 | **82.08** | **81.76** | 81.73 | 81.81 | 81.60 | 81.51 | 81.47 | 81.05 |

Table 8: Test accuracy (Ensemble, Average and Greedy soups) on CIFAR-10 with VGG-11

| | Baseline | | | PAPA | | | PAPA-all | | | PAPA-2 | | |
|---|---|---|---|---|---|---|---|---|---|---|---|---|
| | | Soups | | | Soups | | | Soups | | | Soups | |
| | Ens | Avg | Greed | Ens | Avg | Greed | Ens | Avg | Greed | Ens | Avg | Greed |
| # of models | $p$ | 1 | 1 | $p$ | 1 | 1 | $p$ | 1 | 1 | $p$ | 1 | 1 |
| **No data augmentation or regularization** | | | | | | | | | | | | |
| $p = 3$ | **93.32** | 86.90 | 92.10 | 92.55 | 92.55 | 92.59 | 92.97 | **92.98** | **93.03** | 92.85 | 92.86 | 92.85 |
| $p = 5$ | **93.67** | 82.94 | 92.47 | 93.06 | **93.07** | **93.09** | 92.90 | 92.90 | 92.96 | 92.89 | 92.91 | 92.91 |
| $p = 10$ | **93.80** | 53.34 | 92.11 | 93.24 | **93.26** | **93.21** | 92.80 | 92.81 | 92.86 | 92.86 | 92.85 | 92.78 |
| **With random data augmentations** | | | | | | | | | | | | |
| $p = 3$ | 94.52 | 85.94 | 93.69 | **94.61** | **94.45** | **94.45** | 94.32 | 94.35 | 94.33 | 94.15 | 94.13 | 94.10 |
| $p = 5$ | **94.96** | 81.75 | 93.98 | 94.62 | **94.78** | **94.69** | 94.51 | 94.49 | 94.27 | 94.69 | 94.64 | 94.50 |
| $p = 10$ | **95.22** | 9.49 | 94.11 | 94.95 | **94.82** | **94.61** | 94.37 | 94.35 | 94.48 | 94.61 | 94.47 | 94.48 |

Table 9: Test accuracy (Ensemble, Average and Greedy soups) on CIFAR-10 with ResNet-18

| | Baseline | | | PAPA | | | PAPA-all | | | PAPA-2 | | |
|---|---|---|---|---|---|---|---|---|---|---|---|---|
| | | Soups | | | Soups | | | Soups | | | Soups | |
| | Ens | Avg | Greed | Ens | Avg | Greed | Ens | Avg | Greed | Ens | Avg | Greed |
| # of models | $p$ | 1 | 1 | $p$ | 1 | 1 | $p$ | 1 | 1 | $p$ | 1 | 1 |
| **No data augmentation or regularization** | | | | | | | | | | | | |
| $p = 3$ | **96.00** | 76.11 | 95.47 | 95.97 | **95.97** | **95.96** | 95.75 | 95.76 | 95.70 | 95.72 | 95.72 | 95.73 |
| $p = 5$ | **96.30** | 30.98 | 95.59 | 96.16 | **96.17** | **96.14** | 95.88 | 95.88 | 95.86 | 95.85 | 95.83 | 95.86 |
| $p = 10$ | **96.41** | 10.00 | 95.28 | 95.91 | 95.90 | 95.87 | 95.74 | 95.73 | 95.78 | 96.04 | **96.04** | **96.00** |
| **With random data augmentations** | | | | | | | | | | | | |
| $p = 3$ | **97.32** | 59.91 | 97.05 | 97.31 | **97.30** | 97.16 | 97.04 | 96.98 | 96.94 | 97.14 | 97.12 | **97.17** |
| $p = 5$ | **97.44** | 37.87 | 97.02 | 97.46 | **97.37** | **97.29** | 97.17 | 97.17 | 97.09 | 97.18 | 97.15 | 97.09 |
| $p = 10$ | **97.52** | 11.20 | 96.96 | 97.45 | **97.48** | 97.14 | 97.29 | 97.32 | 97.22 | 97.12 | 97.18 | **97.23** |

### A.8 Image segmentation results

The results for image segmentation on Vaihingen are shown below. We see that for most classes and the average, the F1-score is slightly higher with PAPA variants. However for cars, PAPA variants do slightly worse.

Table 10: Test Accuracy and F1-scores on Vaihingen image segmentation without data augmentation or regularization

| | Roads | Buildings | F1-score Vegetation | Trees | Cars | Clutter | Average F1 |
|---|---|---|---|---|---|---|---|
| **Baseline** ($p = 8$) | | | | | | | |
| Population mean | 88.80 | 91.27 | 80.53 | 87.85 | **79.72** | 29.15 | 76.22 |
| AvgSoup | 43.32 | 0.00 | 0.00 | 0.00 | 0.00 | 0.00 | 7.22 |
| GreedySoup | 89.21 | 91.75 | **80.95** | 87.84 | 78.79 | 29.01 | 76.25 |
| **PAPA-all** ($p = 8$) | | | | | | | |
| Population mean | 88.58 | 91.05 | 80.64 | 87.82 | 79.28 | 29.89 | 76.21 |
| AvgSoup | 88.57 | 91.05 | 80.70 | 87.85 | 79.29 | **29.90** | 76.23 |
| GreedySoup | 88.64 | 91.07 | 80.75 | 87.85 | 79.43 | 29.87 | 76.27 |
| **PAPA-2** ($p = 8$) | | | | | | | |
| Population mean | 89.34 | 91.91 | 80.73 | 87.89 | 79.14 | 29.50 | 76.41 |
| AvgSoup | 89.36 | 91.93 | 80.79 | **87.92** | 79.17 | 29.51 | **76.45** |
| GreedySoup | **89.40** | **92.03** | 80.83 | 87.90 | 79.12 | 29.37 | 76.44 |

Table 11: Test Accuracy and F1-scores on Vaihingen image segmentation with a random draw from: Mixup $\alpha \in \{0, 0.2, 0.4\}$, Label smoothing $\alpha \in \{0, 0.05, 0.1\}$

| | Roads | Buildings | F1-score Vegetation | Trees | Cars | Clutter | Average F1 |
|---|---|---|---|---|---|---|---|
| **Baseline** ($p = 8$) | | | | | | | |
| Population mean | 88.74 | 91.43 | 80.72 | 87.93 | **80.46** | 25.48 | 75.79 |
| AvgSoup | 43.32 | 0.00 | 0.00 | 0.00 | 0.00 | 0.00 | 7.22 |
| GreedySoup | 43.32 | 0.00 | 0.00 | 0.00 | 0.00 | 0.00 | 7.22 |
| **PAPA-all** ($p = 8$) | | | | | | | |
| Population mean | 88.71 | 91.34 | 80.85 | 87.92 | 79.36 | 26.63 | 75.80 |
| AvgSoup | 88.72 | 91.33 | 80.89 | 87.94 | 79.44 | **29.14** | **76.24** |
| GreedySoup | **88.83** | **91.58** | **80.94** | 87.96 | 79.53 | 22.90 | 75.29 |
| **PAPA-2** ($p = 8$) | | | | | | | |
| Population mean | 88.79 | 91.30 | 80.73 | 87.97 | 77.87 | 27.12 | 75.63 |
| AvgSoup | 88.78 | 91.33 | 80.83 | 88.02 | 77.95 | 27.59 | 75.75 |
| GreedySoup | 88.68 | 91.29 | 80.93 | **88.03** | 77.89 | 28.66 | 75.91 |

### A.9 Comparing PAPA variants to ESGD

Table 12: Comparing PAPA variants to ESGD on CIFAR-10

| Method | p | mean [min, max] |
|---|---|---|
| Baseline population (reported in Cui et al. (2018)) | 128 | 91.76 [91.31, 92.10] |
| ESGD (reported in Cui et al. (2018)) | 128 | **92.48** [91.90, 92.57] |
| Baseline population (our re-implementation) | 3 | 91.55 [91.48, 91.67] |
| Baseline population (our re-implementation) | 5 | 91.61 [91.24, 91.83] |
| Baseline population (our re-implementation) | 10 | 91.42 [91.04, 91.82] |
| PAPA | 3 | 92.57 [92.22, 92.99] |
| PAPA-all | 3 | 92.72 [92.45, 93.19] |
| PAPA-2 | 3 | 92.47 [92.25, 92.87] |
| PAPA | 5 | 92.91 [92.40, 93.36] |
| PAPA-all | 5 | 92.48 [91.98, 92.95] |
| PAPA-2 | 5 | 92.01 [91.71, 92.46] |
| PAPA | 10 | **93.00** [92.65, 93.31] |
| PAPA-all | 10 | 92.36 [92.05, 92.78] |
| PAPA-2 | 10 | 92.07 [91.73, 92.58] |

evolutionary stochastic gradient descent (ESGD) (Cui et al., 2018) is a method that leverages a population of networks through a complicated genetic algorithm. In this section, we compare PAPA variants to ESGD. Since ESGD did not release their code as open source, direct comparison is difficult. Rather than re-implementing ESGD from scratch, we replicate the same image recognition experiment used by Cui et al. (2018) and compare PAPA approaches to ESGD in this setting. From the paper, the experiments were done with CIFAR-10, using the same data processing as our main experiments using a different schedule (160 epochs, learning starts at 0.1 and is decayed by a scaling of 0.1 at 81 epochs and again at 122 epochs). The authors do not mention what batch size they used, but having tried 32, 64, and 128, 128 gives us numbers closest to the one from their paper; thus, we use a batch size of 128. We do not report soups since we can only compare the results with respect to the mean, min, and max since these were the metrics reported in Cui et al. (2018). The architecture used is the same ResNet-20 that they used. Note that we used $\alpha_{\mathrm{papa}} = 0.95$ in this setting.

The results are shown in the table below. Our baseline population replication has similar test accuracy to the one reported by Cui et al. (2018). PAPA variants attain higher mean accuracy than ESGD using 3-10 models instead of 128.

### A.10 Logistic regression (single-layer)

This section compares baseline to PAPA variants when doing simple logistic regression on the Optical Handwritten Digits Dataset (Dua and Graff, 2017). We use SGD with batch-size 1, a constant learning rate of 0.1 for 10 epochs. PAPA-all averages at every epoch. We return the average model soup and mean accuracy at the end of training. We use $p = 10$ and average the results over 10 random runs.

For the baseline, we obtain a mean accuracy of 92.76% and average soup accuracy of 93.62%. As can be seen, the average soup works very well. This suggests that alignment or PAPA variants are unneeded in the single-layer setting. For PAPA-all, we obtain a mean accuracy of 91.34% and average soup accuracy of 90.37%. Thus PAPA-all performs worse than the baseline in this scenario.

These results suggest that model soups can already work well without alignment in the single-layer setting and that PAPA benefits are specific to deep neural networks.

### A.11 Combining PAPA variants and SWA

This section shows results with stochastic weight averaging (SWA) (Izmailov et al., 2018) to demonstrate that averaging over different models (PAPA avariants) provides additional benefits that averaging over a single trajectory (SWA) does not provide.

We train ResNet-18 models on CIFAR-100 with a population of $p = 5$ networks. We use a multistep learning rate schedule (start at 0.1, decay by 10 at 150 and 225 epochs) and varying data augmentations and

regularizations (random draw from Mixup $\alpha \in [0,1]$ and Label smoothing $\alpha \in [0,0.1]$). For SWA, we follow the training protocol of Izmailov et al. (2018), which consists of training for 75% of the regular training time without SWA (225 epochs) followed by an extra 75% of the regular training time with SWA (for a total of 550 epochs). We show results on different choices of learning rate schedules. We compared the baseline, PAPA-all only applied before SWA starts (first 75% of regular training), and PAPA-all applied during the whole training process. Results are shown in Table 13.

Table 7 shows that the mean accuracy for baseline models is 79.35%. Using SWA, baseline networks attain a mean accuracy of 80.77%, thus 1.42% higher. Table 7 shows that the mean accuracy for PAPA-all models is 80.21%. By using SWA, PAPA-all networks attain a mean accuracy of 81.88%, thus 1.67% higher. Hence, PAPA-all networks with SWA have a mean accuracy that is 1.11% higher than baseline models with SWA.

Consequently, SWA is highly beneficial, and PAPA-all provides significant additional benefits beyond those made by SWA. This demonstrates that SWA and PAPA variants are different methods that can complement one another.

Table 13: Test accuracy (Ensemble, Average and Greedy soups) on CIFAR-100 with ResNet-18 using SWA on a population of $p = 5$ networks with varying data augmentations and regularizations

| | | Baseline (SWA) | | | PAPA-all | | | PAPA-all | | |
|---|---|---|---|---|---|---|---|---|---|---|
| Averaging before SWA | | | | | ✓ | | | ✓ | | |
| Averaging during SWA | | | | | | | | ✓ | | |
| | | Population $p$ models | Soups 1 model | | Population $p$ models | Soups 1 model | | Population $p$ models | Soups 1 model | |
| Schedule | learning rate | Mean | Avg | Greed | Mean | Avg | Greed | Mean | Avg | Greed |
| Linear | 0.05 | 80.62 | 14.21 | 71.44 | 81.08 | 79.16 | 70.96 | **81.54** | **80.53** | **71.66** |
| Cosine | 0.05 | 80.56 | 15.04 | **71.98** | 81.21 | 79.81 | 70.62 | **81.88** | **80.41** | 71.02 |
| Linear | 0.01 | 80.70 | 8.17 | 76.72 | 81.16 | 81.19 | 76.87 | **81.86** | **81.25** | **76.94** |
| Cosine | 0.01 | 80.77 | 12.22 | 76.72 | 81.62 | **81.87** | 77.73 | **81.80** | 81.30 | **79.98** |

## A.12  Ablation

We conduct an ablation with $p = 5$, multistep learning rate schedule (start at 0.1, decay by 10 at 150 and 225 epochs) and varying data augmentations and regularizations (random draw from Mixup $\alpha \in [0,1]$ and Label smoothing $\alpha \in [0,0.1]$) on CIFAR-100.

We test various options: changing the averaging method, using the same initializations, using no REPAIR, using permutation alignment, only averaging during some training periods, adding aspects of Genetic Algorithms (GAs) from ESGD (Cui et al., 2018), such as random mutations ($\mathcal{N}(0, (\frac{0.01}{g})^2)$, where $g$ is the number of generations passed), tournament selection, elitist selection (making $6p$ children, selecting the top 60% of the population and randomly selecting the rest). For PAPA, we test different choices of $\alpha_{papa}$.

We observe that averaging two models (PAPA-2) is the best strategy, followed closely by averaging the weights of all models (PAPA-all).

We generally see that averaging at different frequencies, using the same initialization for every model, removing REPAIR, using feature-matching permutation alignment (Ainsworth et al., 2023), and adding evolutionary elements decrease performance; the only exception is with PAPA-all where averaging every ten epochs performs better.

We investigate averaging only during certain epochs instead through the whole training. We observe that mean accuracy decreases the more we reduce the averaging time-period. Interestingly, average soups (but not greedy soups) are better when averaging only for the first half of training; however, that difference is insignificant for PAPA-all.

Regarding PAPA, we test it on slightly different choices of $\alpha_{papa}$ while using it after every SGD step; we see that $\alpha_{papa} = 0.999$ is optimal. We arbitrarily chose to amortize it over 10 steps ($\alpha_{papa} = 0.999^{10} \approx 0.99$) to minimize the slowdown from having to calculate the population average of the weights; performance is better with this choice. We did not explore any other settings.

Table 14: Ablation (multistep schedule with Mixup and Label smoothing) between different averaging methods

| Method | mean [min, max] | AvgSoup | GreedySoup |
|---|---|---|---|
| [PAPA-all] average ($w_i = \frac{1}{p}$ $\forall i$) | 80.21 [80.05, 80.46] | 80.52 | 80.24 |
| [PAPA-2] pair-half ($w_1 = w_2 = \frac{1}{2}, w_i = 0$ $\forall i \geq 3$) | **80.31** [80.12, 80.59] | **80.57** | **80.56** |
| pair-75 ($w_1 = \frac{3}{4}, w_2 = \frac{1}{4}, w_i = 0$ $\forall i \geq 3$) | 79.58 [79.32, 80.02] | 79.84 | 79.32 |
| many-half ($w_1 = \frac{1}{2}, w_i = \frac{1}{2(p-1)}$ $\forall i \geq 2$) | 79.92 [79.64, 80.05] | 80.12 | 79.94 |
| many-75 ($w_1 = \frac{3}{4}, w_i = \frac{1}{4(p-1)}$ $\forall i \geq 2$) | 68.62 [68.14, 69.22] | 29.70 | 69.22 |
| GreedySoup | 79.95 [79.61, 80.23] | 80.24 | 79.99 |
| [Baseline] no averaging | 79.35 [78.97, 79.73] | 9.26 | 79.73 |

Table 15: Ablation (multistep schedule with Mixup and Label smoothing) when using all-models averaging (PAPA-all)

| Method | mean [min, max] | AvgSoup | GreedySoup |
|---|---|---|---|
| PAPA-all | 80.21 [80.05, 80.46] | 80.52 | 80.24 |
| averaging every $k = 1$ epochs | 79.90 [79.72, 80.28] | 80.16 | 79.76 |
| averaging every $k = 10$ epochs | **80.44** [79.98, 80.67] | 80.90 | **80.72** |
| same initialization | 79.96 [79.57, 80.20] | 80.32 | 80.04 |
| REPAIR 1-iter | 79.98 [79.57, 80.40] | 80.38 | 80.40 |
| no-REPAIR | 79.56 [79.29, 79.79] | 79.71 | 79.69 |
| feature-matching permutation alignment | 79.30 [78.93, 79.48] | 79.31 | 79.35 |
| mutations | 80.20 [79.84, 80.53] | 80.41 | 80.41 |
| average only from epochs 0 to 75 | 79.55 [79.06, 80.07] | 79.80 | 79.89 |
| average only from epochs 0 to 150 | 79.63 [79.14, 80.03] | **80.99** | 79.60 |
| average only from epochs 0 to 225 | 79.80 [79.30, 80.15] | 80.53 | 80.35 |
| average only from epochs 225 to 300 | 64.90 [64.31, 65.26] | 66.32 | 66.06 |
| average only from epochs 150 to 300 | 74.73 [74.48, 74.97] | 74.98 | 74.96 |
| average only from epochs 75 to 300 | 79.91 [79.63, 80.20] | 80.25 | 79.98 |

Table 16: Ablation (multistep schedule with Mixup and Label smoothing) when using two-models averaging (PAPA-2)

| Method | mean [min, max] | AvgSoup | GreedySoup |
|---|---|---|---|
| PAPA-2 | **80.31** [80.12, 80.59] | 80.57 | **80.56** |
| averaging every $k = 1$ epochs | 79.72 [79.48, 79.79] | 79.87 | 79.74 |
| averaging every $k = 10$ epochs | 79.93 [79.59, 80.19] | 80.18 | 80.08 |
| same initialization | 79.92 [79.78, 80.09] | 80.19 | 80.16 |
| REPAIR 1-iter | 80.23 [79.89, 80.49] | 80.38 | 80.27 |
| no-REPAIR | 80.02 [79.73, 80.45] | 80.49 | 79.73 |
| feature-matching permutation alignment | 79.83 [79.41, 80.17] | 80.27 | 79.52 |
| mutations | 80.00 [79.76, 80.29] | 80.14 | 80.29 |
| pair-half with tournament selection | 79.71 [79.54, 80.00] | 80.01 | 79.95 |
| pair-half, tournament, elitist | 79.39 [79.12, 79.59] | 1.0 | 79.59 |
| pair-half, tournament, elitist, mutations | 79.49 [78.59, 80.01] | 1.0 | 80.01 |
| tournament, elitist, mutations, no-REPAIR | 79.44 [78.83, 79.91] | 1.0 | 79.30 |
| average only from epochs 0 to 75 | 79.57 [79.18, 79.82] | 79.50 | 79.62 |
| average only from epochs 0 to 150 | 79.64 [79.25, 80.11] | **81.17** | 80.04 |
| average only from epochs 0 to 225 | 79.78 [79.29, 80.32] | 80.36 | 80.32 |
| average only from epochs 225 to 300 | 76.12 [75.86, 76.40] | 76.43 | 76.33 |
| average only from epochs 150 to 300 | 78.03 [77.80, 78.32] | 78.01 | 78.05 |
| average only from epochs 75 to 300 | 80.20 [79.85, 80.36] | 80.46 | 80.54 |

Table 17: Ablation (multistep schedule with Mixup and Label smoothing) when using PAPA

| Method | mean [min, max] | AvgSoup | GreedySoup |
|---|---|---|---|
| PAPA | **79.85** [78.71, 80.46] | 80.30 | 80.20 |
| same initialization | 79.81 [78.63, 80.33] | 80.37 | **80.57** |
| feature-matching permutation alignment | 79.12 [78.17, 79.59] | 79.78 | 79.82 |
| mutations | 79.70 [78.43, 80.26] | **80.40** | 79.84 |
| average only from epochs 0 to 75 | 78.51 [77.73, 78.78] | 79.28 | 78.78 |
| average only from epochs 0 to 150 | 78.91 [77.98, 79.41] | 80.20 | 79.92 |
| average only from epochs 0 to 225 | 79.36 [77.98, 79.89] | 80.16 | 79.66 |
| average only from epochs 225 to 300 | 77.61 [76.51, 78.47] | 1.00 | 78.47 |
| average only from epochs 150 to 300 | 76.22 [75.58, 76.69] | 1.06 | 76.57 |
| average only from epochs 75 to 300 | 78.99 [77.84, 79.52] | 79.59 | 79.63 |
| every 1 SGD steps with $\alpha_{papa} = 0.9999$ | 78.43 [77.56, 79.08] | 1.00 | 78.58 |
| every 1 SGD steps with $\alpha_{papa} = 0.9995$ | 79.11 [78.10, 79.59] | 79.76 | 79.59 |
| every 1 SGD steps with $\alpha_{papa} = 0.999$ | 79.22 [77.61, 80.02] | 79.82 | 79.95 |
| every 1 SGD steps with $\alpha_{papa} = 0.995$ | 78.97 [77.60, 79.43] | 79.67 | 79.70 |
| every 1 SGD steps with $\alpha_{papa} = 0.99$ | 78.07 [76.77, 78.66] | 78.53 | 78.46 |
| every 1 SGD steps with $\alpha_{papa} = 0.95$ | 78.57 [77.37, 79.01] | 78.92 | 79.12 |
| every 1 SGD steps with $\alpha_{papa} = 0.9$ | 78.87 [77.86, 79.21] | 79.23 | 79.35 |
| every 1 SGD steps with $\alpha_{papa} = 0.5$ | 78.34 [77.82, 78.62] | 78.61 | 78.32 |
| every 1 SGD steps with $\alpha_{papa} = 0.1$ | 78.35 [77.86, 78.69] | 78.53 | 78.59 |
| every 10 SGD steps with $\alpha_{papa} = 0.999$ | 78.12 [76.96, 78.73] | 1.00 | 78.45 |
| every 10 SGD steps with $\alpha_{papa} = 0.995$ | 79.36 [77.87, 79.97] | 80.20 | 80.13 |
| every 10 SGD steps with $\alpha_{papa} = 0.99$ | 79.39 [78.38, 80.09] | 79.99 | 80.28 |
| every 10 SGD steps with $\alpha_{papa} = 0.95$ | 78.72 [77.75, 79.43] | 79.18 | 79.48 |
| every 10 SGD steps with $\alpha_{papa} = 0.9$ | 78.47 [77.39, 78.92] | 78.83 | 78.77 |
| every 10 SGD steps with $\alpha_{papa} = 0.5$ | 78.71 [77.47, 79.17] | 79.15 | 79.13 |
| every 10 SGD steps with $\alpha_{papa} = 0.1$ | 78.27 [77.13, 78.71] | 78.63 | 78.24 |
| every 50 SGD steps with $\alpha_{papa} = 0.995$ | 78.38 [77.65, 79.09] | 1.09 | 79.09 |
| every 50 SGD steps with $\alpha_{papa} = 0.99$ | 78.05 [77.01, 78.51] | 71.12 | 78.15 |
| every 50 SGD steps with $\alpha_{papa} = 0.95$ | 79.12 [77.74, 79.67] | 79.57 | 78.88 |
| every 50 SGD steps with $\alpha_{papa} = 0.9$ | 79.48 [78.42, 79.96] | 79.94 | 79.96 |
| every 50 SGD steps with $\alpha_{papa} = 0.5$ | 78.29 [76.81, 78.80] | 78.83 | 79.03 |
| every 50 SGD steps with $\alpha_{papa} = 0.1$ | 78.67 [77.41, 79.15] | 79.03 | 79.03 |

### A.13 Greedy soups

We provide the number of models included in the baseline greedy soups. As can be seen, with pre-training, models are not amenable to averaging, and thus only the best model is used. Meanwhile, for fine-tuning, models are more amenable to averaging since they do not move too far from the common initialization, and thus multiple models are combined.

Table 18: Number of models included in the greedy soups of the main experiments (Table 1)

| | Baseline |
|---|---|
| | GreedySoup |
| **CIFAR-10** ($n_{epochs} = 300$, $p = 10$) | |
| VGG-11 | 1 (out of 10) |
| ResNet-18 | 1 (out of 10) |
| **CIFAR-100** ($n_{epochs} = 300$, $p = 10$) | |
| VGG-16 | 1 (out of 10) |
| ResNet-18 | 1 (out of 10) |
| **Imagenet** ($n_{epochs} = 90$, $p = 3$) | |
| ResNet-50 | 1 (out of 3) |
| **Fine-tuning on CIFAR-100** | |
| EffNetV2-S | 2 (out of 2) |
| EVA-02-Ti | 3 (out of 4) |
| ConViT-Ti | 3 (out of 5) |

