# OpenReview forum: "PopulAtion Parameter Averaging (PAPA)"
_TMLR — Accepted by TMLR_

### Review · Reviewer_Fjge · 2023-11-27

**Summary Of Contributions:**

This paper introduces PopulAtion Parameter Averaging (PAPA), a method aimed at enhancing the generalization performance of neural networks. PAPA operates by utilizing a diverse population of models, each trained on the same dataset but under varying conditions such as data orderings, augmentations, and regularization techniques. The process involves three key steps: i) Training multiple models in parallel, each subjected to different training conditions, ii) Periodically performing weight averaging across these models, which effectively nudges each model towards a common population average, iii) Culminating in the creation of a single "model soup" by averaging the final weights of all models in the population.

The effectiveness of PAPA is empirically validated through a series of experiments conducted on CIFAR and ImageNet for classification tasks, as well as on an image segmentation task. The results from these experiments show that PAPA yields improved accuracy compared to baseline methods, including traditional model ensembling and the model soup technique.

**Audience:**

No

**Claims And Evidence:**

No

**Requested Changes:**

More experiments are definitely needed, for modern pre-trained models (e.g., CLIP, DINOv2) on ImageNet.

This paper lacks insights about the algorithm proposal. The authors should offer interesting insights about why PAPA can improve over Model Soup.

**Strengths And Weaknesses:**

## Strengths

+ The method is simple to implement, requiring only period weight averaging of independently trained networks.

## Weaknesses

+ [Not an Original Proposed Algorithm] The PAPA method described in this paper is not an original algorithm. It closely resembles FedAvg, with the only distinction being that each subprocess in PAPA is trained on the entire dataset rather than a subset. While TMLR may not prioritize novelty extensively, I believe an algorithm paper should still propose novel approaches instead of slightly modifying existing algorithms.

+ [Incremental Contribution & Lack of Insights] The contribution of PAPA, compared to Model Soup, appears marginal and incremental. The Model Soup algorithm, while not complex, gained significant attention because it highlighted an unexpected benefit of weight-space averaging across multiple independent runs. However, PAPA only introduces a minor modification to Model Soup and fails to provide a convincing explanation for the efficacy of this change. Overall, the paper does not offer new insights to the community.

+ [Experiments Too Limited] The Model Soup paper concentrated on fine-tuning pretrained vision models (e.g., CLIP, ALIGN, ViT) for ImageNet, providing a wealth of experimental results. In contrast, the experiments in the PAPA paper are primarily conducted on CIFAR. The only ImageNet experiment involves ResNet with supervised training from scratch. The absence of fine-tuning experiments with modern pretrained models on ImageNet makes the results less compelling. Further, the experiments of fine-tuning SoTA pre-trained models (EffNetV2, EVA-02) show a very tiny improvement over the Model Soup baseline, indicating that the performance gain of PAPA may diminish for strong pretrained models -- I think this will still be the case for fine-tuning SoTA pre-trained models (e.g., CLIP, DINOv2) on ImageNet.

---

> ### Author Response · Authors · 2024-01-19
> **Response to Reviewer Fjge**
>
> Thank you for your comments and suggestions. Below, we clarify the concerns about the lack of novelty that you mentioned, but we want to point out that the main guidelines for TMLR are about whether claims are well supported and not about novelty.
>
> 1) [Not an Original Proposed Algorithm] The PAPA method described in this paper is not an original algorithm. It closely resembles FedAvg, with the only distinction being that each subprocess in PAPA is trained on the entire dataset rather than a subset.
>    - There are more differences between our method and FedAvg. In addition to the training on full data (to perform at least as well as full-data training rather than performing worse like in FedAvg),  we also use different data augmentations per model (to enhance diversity and thus gain generalization when merging models) and a continuous push toward the average using an Exponential Moving Average instead of occasional averaging (for a frequent slight push toward the average which usually leads to better generalization). Our approach is novel given that these differences are non-trivial and required to go from ‘FedAvg performing slightly worse than regular full-data training’ to ‘PAPA generalizing better than full-data training’.
>
> 2) [Incremental Contribution & Lack of Insights] PAPA only introduces a minor modification to Model Soup.
>    - Our paper shows that although greedy model soups provide a slight gain in generalization, it is solely because they choose the single model with the best validation loss from the multiple models. We find that model soups never choose more than one model at a time (which is now mentioned in a footnote to the main table). This is because the models are not amenable to averaging. Our method renders the models amenable to averaging by frequently pushing the models slightly toward their average. Because of this and the diversity in our models (due to the varying data augmentations), our method can average all models for the best gain in generalization. Thus, PAPA cannot be considered incremental over model soups when it solves a fundamental problem with model soup in the specific case of pretraining.
>
> 3) [Experiments Too Limited] Experiments of fine-tuning SoTA pre-trained models (EffNetV2, EVA-02) show a very tiny improvement over the Model Soup baseline, I think this will still be the case for fine-tuning SoTA pre-trained models (e.g., CLIP, DINOv2) on ImageNet.
>    - Our experiments mainly focus on pretraining. We agree that fine-tuning CLIP and DINOv2 models would lead to worse results with PAPA than with model soup. This is because CLIP and DINOv2 normally only fine-tune the last layer (i.e., linear probing), and we have already reported worse results for PAPA for the single-layer logistic regression case. We should have been clearer about the fact that our method is made to fix a problem with pretraining. We only found PAPA to be mildly beneficial when fine-tuning all parameters. However, one should not expect any benefit when fine-tuning the last layer. Given the importance of these aspects, we now mention this immediately before the conclusion in a new discussion section:
> > During pretraining, greedy model soups only used a single model and performed worse than PAPA variants. During fine-tuning, greedy model soups used multiple models. In the case of training a single layer, our experiments showed that PAPA performed worse than greedy soups. The conclusion from these experiments is the following:
> 	> - During pretraining, PAPA ensures that the models stay similar enough so that model averaging provides equal or greater generalization. Meanwhile, model soups will only choose the best network because the models are not amenable to averaging, and doing so would lead to a massive drop in generalization.
> 	> - During fine-tuning from a pre-trained model, models are amenable to averaging, and thus, PAPA is unnecessary over model soups. We still found small benefits of PAPA over model soups in the case of fine-tuning all parameters. However, this is because we fine-tuned all parameters. When training a single layer (as is the case for linear probing), PAPA is unnecessary because averaging the weights of a single linear layer already works since the permutations are perfectly aligned.
>
> 4) The authors should offer interesting insights about why PAPA can improve over Model Soup.
>    - As mentioned in the point above (#3), we added a crucial missing discussion before the conclusion on why PAPA is needed when pretraining. In short, this discussion highlights that model soups only choose a single model as they are not amenable to averaging different models. It's only through regular averaging (PAPA-all) or pushing the weights frequently toward the average (PAPA) that we can keep the models similar enough so that averaging does not reduce performance.

---

### Review · Reviewer_J8GN · 2023-12-01

**Summary Of Contributions:**

The authors consider an approach for averaging weights from an ensemble for an improvement in performance. They showed that their approach based on the mixing of average weights and weights of a single model improves the final results for considered experimental scenarios.

**Audience:**

Yes

**Broader Impact Concerns:**

-

**Claims And Evidence:**

Yes

**Requested Changes:**

- Add more baselines to the main text of the article and move these results from the appendix
- I feel that the size of Figure 1 and Algorithm 1 can be reduced, as they now occupy large parts of the article with relatively minor contribution
- Mention that DART is accepted to CVPR 2023 and is no longer an arxiv preprint

**Strengths And Weaknesses:**

Strengths:
- The idea is easy to implement and nice
- We see the mentioned improvement in quality in the provided experiments. The experiments consider major benchmarks close to those adopted in other papers. The hyperparameters are also close.
- The authors show that a concurrent work DART provides similar results for a variant of their

Weaknesses:
- The choice of hyperparameters for the procedure should be more straightforward and easier to spot in the article
- The authors mention a concurrent work, DART, and it is very similar to the ideas presented in the PAPA work

---

> ### Author Response · Authors · 2024-01-19
> **Response to Reviewer J8GN**
>
> Thank you for your comments and suggestions. We address your concerns and requested changes below:
>
> 1) The choice of hyperparameters for the procedure should be more straightforward and easier to spot in the article
>    - We now mention the hyperparameters used in the experiments section.
> 2) Add more baselines to the main text of the article and move these results from the appendix
>    - We now include the sections on “Comparing PAPA variants to baseline models trained for p times more epochs” and “Comparing PAPA to DART” in the main text.
> 3) I feel that the size of Figure 1 and Algorithm 1 can be reduced, as they now occupy large parts of the article with relatively minor contribution
>    - We reduced the size of Figure 1 and trimmed the size of Algorithm 1.
> 4) Mention that DART is accepted to CVPR 2023 and is no longer an arxiv preprint
>    - We now mention this in the paper and we also updated the reference to refer to CVPR.

---

### Review · Reviewer_2qLT · 2024-01-13

**Summary Of Contributions:**

Previous work has established that, under the right circumstances, the weights of different neural networks can be averaged together to produce a new network that performs better. This submission proposes a family of methods to train multiple networks simultaneously for subsequent weight averaging where each network is frequently interpolated (PAPA) or occasionally replaced (PAPA-2/PAPA-all) with an average. This average can be the average of all networks' weights or, in the case of PAPA-2, the average of a random pair of weights. Optionally, REPAIR can be applied to adjust batch norm activations. The submission presents experiments in a variety of settings, including training from scratch on CIFAR-10, CIFAR-100, ImageNet; fine-tuning on CIFAR-100; and training from scratch on the ISPRS Vaihingen remote sensing segmentation dataset. Results show that PAPA generally outperforms other methods, but PAPA-all and PAPA-2 are typically at least a bit better than a greedy soup baseline.

**Audience:**

Yes

**Broader Impact Concerns:**

No concerns.

**Claims And Evidence:**

Yes

**Requested Changes:**

See the first two points under "Weaknesses." The third point doesn't necessarily need to be addressed for this paper to meet TMLR publication criteria.

**Strengths And Weaknesses:**

Strengths:
- The submission explores a wide variety of different settings. The experiments and baselines appear to be properly performed.
- The methodology is clearly described, with full details of the proposed method in the main text and complete procedures for all of the methods the paper builds upon in either the main text or appendix.
- Though not a requirement for TMLR, the proposed method provides a small performance improvement.

Weaknesses:
- It isn't immediately clear whether the proposed method outperforms other ways that the same computational budget could be allocated to could be allocated to improve performance. Conclusively demonstrating that the proposed method outperforms everything else is hard, and not necessary for TMLR as long as the authors are careful about their claims. There is one claim that I think might be problematic: The submission states that "PAPA obtains an accuracy of 78.36% on ImageNet. Reaching such accuracy with ResNet-50 usually requires large batch sizes...or long training." I'm not aware of a better number off the top of my head, but I'm not convinced that obtaining this result _requires_ a large batch size; I think it may just be that people who optimize ImageNet models generally use big batch sizes so they can train faster. [1] uses a batch size of 256 and primarily explores an architecture that is a bit different from standard ResNet-50, but subtracting out the gain they get from changing the architecture, their accuracy with cosine decay, label smoothing, and mixup would be around what is reported here.
- In Table 1, I suspect that the "GreedySoup baseline" is just an individually trained model for the train-from-scratch settings, since souping wouldn't be expected to work here. It might help the reader to state this explicitly, perhaps in the caption or using an asterisk.
- Between PAPA-all and PAPA-2, it isn't exactly clear which method one would want to use in practice. This doesn't affect the submission's suitability for publication in TMLR, but its message may be somewhat more diluted than if it focused primarily on one of the two methods in the main text and deferred the other to the appendix.

Minor:
- Proposition 1 is more of a definition than a proposition.

References:\
[1] He, T., Zhang, Z., Zhang, H., Zhang, Z., Xie, J., & Li, M. (2019). Bag of tricks for image classification with convolutional neural networks. In Proceedings of the IEEE/CVF conference on computer vision and pattern recognition (pp. 558-567).

---

> ### Author Response · Authors · 2024-01-19
> **Response to Reviewer 2qLT**
>
> Thank you for your comments and suggestions. We address your concerns and requested changes below:
>
> 1) It isn't immediately clear whether the proposed method outperforms other ways that the same computational budget could be allocated to improve performance.
>    - We provide experiments comparing training a population of p models using PAPA-all to training a single model for p times more epochs. These experiments show the benefits of using PAPA over a single model longer, given a fixed budget. These experiments were in the Appendix, but as per reviewer J8GN suggestion, we now moved them to the main text.
>
> 2) There is one claim that I think might be problematic: The submission states that "PAPA obtains an accuracy of 78.36% on ImageNet. Reaching such accuracy with ResNet-50 usually requires large batch sizes...or long training." I'm not aware of a better number off the top of my head, but I'm not convinced that obtaining this result requires a large batch size; I think it may just be that people who optimize ImageNet models generally use big batch sizes so they can train faster. [1] uses a batch size of 256 and primarily explores an architecture that is a bit different from standard ResNet-50, but subtracting out the gain they get from changing the architecture, their accuracy with cosine decay, label smoothing, and mixup would be around what is reported here.
>    - We agree that this claim could be potentially debated since the larger batches may only be for efficiency reasons. We revised the text
>       - Previous version: Reaching such accuracy with ResNet-50 usually requires large batch sizes (1024 for 300 epochs, 2048 for 100 epochs) or long training (384 for 600 epochs) (Wightman et al., 2021). Meanwhile, we use only a batch size of 256 for 90 epochs with 3 networks (equivalent in compute to 270 epochs on a single GPU). Thus, PAPA makes it possible to use small batches in order to obtain similar performance to large-batch training.
>       - New version: Previous results with similar accuracy were obtained using large batch sizes (1024 for 300 epochs, 2048 for 100 epochs) or long training (384 for 600 epochs) \citep{wightman2021resnet}. Meanwhile, we use only a batch size of 256 for 90 epochs with 3 networks (equivalent in compute to 270 epochs on a single GPU).
>
>    Let us know if this revised sentence is sufficient to address the problematic claim. We can also remove these sentences completely if preferred.
>
> 3) In Table 1, I suspect that the "GreedySoup baseline" is just an individually trained model for the train-from-scratch settings, since souping wouldn't be expected to work here. It might help the reader to state this explicitly, perhaps in the caption or using an asterisk.
>    - Your assumption is correct. GreedySoup greedily tries to add each model to the soup, and since models are not amenable, it chooses only the single best model (based on the validation set). We show this in Appendix A.13. We added a footnote to “GreedySoup” in the main table to mention this so the reader can know directly from looking at the results.
>
> 4) Between PAPA-all and PAPA-2, it isn't exactly clear which method one would want to use in practice. This doesn't affect the submission's suitability for publication in TMLR, but its message may be somewhat more diluted than if it focused primarily on one of the two methods in the main text and deferred the other to the appendix.
>    - We mention in the conclusion that “PAPA tends to be the best method out of all PAPA variants”. We updated the sentence to be more clear by saying: “In practice, we find that all methods increase generalization, but PAPA tends to perform better than PAPA-all and PAPA-2.” In the abstract, we now also mention the following: “We also propose PAPA variants (PAPA-all, and PAPA-2) that average weights rarely rather than continuously; all methods increase generalization, but PAPA tends to perform best.”
>
> 5) Proposition 1 is more of a definition than a proposition.
>    - Following your suggestion, we made it a Definition.

---

### Decision · Action_Editor_uWtC · 2024-02-29

**Recommendation:** Accept with minor revision

**Comment:**

The paper received one "accept" and one "reject" recommendation, with the third reviewer unfortunately not submitting a recommendation in reasonable time. Based on my reading of the reviews, recommendations, author replies, and the paper, I believe it should be accepted to TMLR.

The main arguments for rejecting the paper are the incremental nature of its contribution, the fact that it does not contain more diverse experiments, and the fact that the improvement over model soups seems small for fine-tuned models. To the best of my understanding these issues are not grounds for rejection from TMLR as long as the claims in the paper are accurate. To improve accuracy and clarity, I request the following minor revisions:
1. In item 3 in the end of the introduction, clarify that these gains are observed when training models from scratch.
2. In the limitations paragraph of Section 6, mention that the benefit of PAPA is smaller when fine-tuning pre-trained models.

**Audience:**

Yes.

**Claims And Evidence:**

The paper proposes a suite of techniques intended to make a population of trained models more amenable to weight averaging. The paper claims the method provides clear improvements on small populations of models on somewhat small-scale image classification tasks, and back these claims with appropriate evidence. The clarity of exposition is fair and was improved thanks to feedback from reviewers. I am requesting a few small edits to further improve the accuracy of the claims in the paper (see comments below).